# GrounDiT: Grounding Diffusion Transformers via Noisy Patch Transplantation

Phillip Y. Lee*    Taehoon Yoon*    Minhyuk Sung

KAIST

{phillip0701,taehoon,mhsung}@kaist.ac.kr

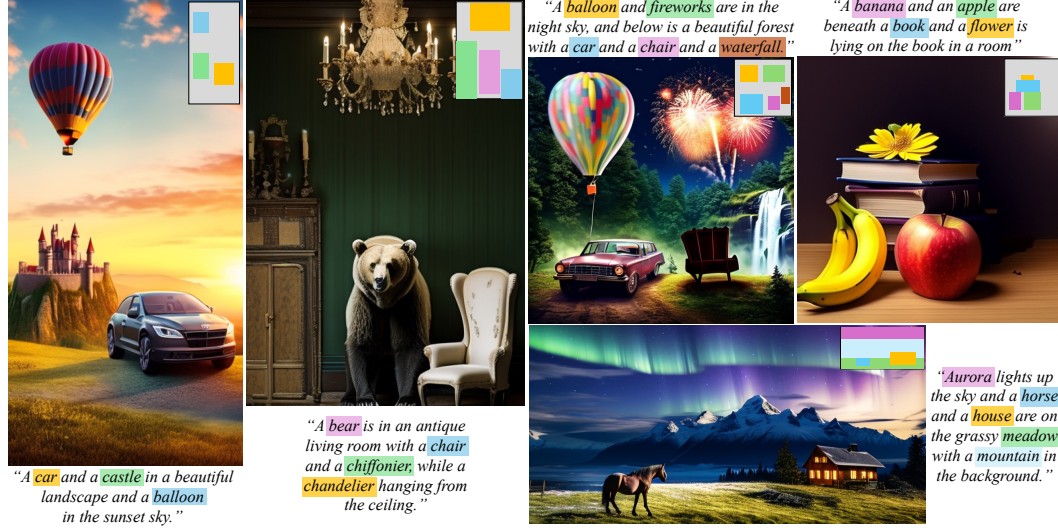

Figure 1: Spatially grounded images generated by our GROUNDIT. Each image is generated based on a text prompt along with bounding boxes, which are displayed in the upper right corner of each image. Compared to existing methods that often struggle to accurately place objects within their designated bounding boxes, our GROUNDIT enables more precise spatial control through a novel noisy patch transplantation mechanism.

## Abstract

We introduce GROUNDIT, a novel training-free spatial grounding technique for text-to-image generation using Diffusion Transformers (DiT). Spatial grounding with bounding boxes has gained attention for its simplicity and versatility, allowing for enhanced user control in image generation. However, prior training-free approaches often rely on updating the noisy image during the reverse diffusion process via backpropagation from custom loss functions, which frequently struggle to provide precise control over individual bounding boxes. In this work, we leverage the flexibility of the Transformer architecture, demonstrating that DiT can generate noisy patches corresponding to each bounding box, fully encoding the target object and allowing for fine-grained control over each region. Our approach builds on an intriguing property of DiT, which we refer to as *semantic sharing*. Due to semantic sharing, when a smaller patch is jointly denoised alongside a generatable-size image, the two become *semantic clones*. Each patch is denoised in its own branch of the generation process and then transplanted into the corresponding region of the original noisy image at each timestep, resulting in robust spatial grounding for each bounding box. In our experiments on the HRS and DrawBench benchmarks, we achieve state-of-the-art performance compared to previous training-free approaches. Project Page: https://groundit-diffusion.github.io/.

---

*Equal contribution.

38th Conference on Neural Information Processing Systems (NeurIPS 2024).

# 1 Introduction

The Transformer architecture [45] has driven breakthroughs across a wide range of applications, with diffusion models emerging as significant recent beneficiaries. Despite the success of diffusion models with U-Net [42] as the denoising backbone [22, 43, 41, 39], recent Transformer-based diffusion models, namely Diffusion Transformers (DiT) [37], have marked another leap in performance. This is demonstrated by recent state-of-the-art generative models such as Stable Diffusion 3 [13] and Sora [6]. Open-source models like DiT [37] and its text-guided successor PixArt-$\alpha$ [8] have also achieved superior quality compared to prior U-Net-based diffusion models. Given the scalability of Transformers, Diffusion Transformers are expected to become the new standard for image generation, especially when trained on an Internet-scale dataset.

With high quality image generation achieved, the next critical step is to enhance user controllability. Among the various types of user guidance in image generation, one of the most fundamental and significant is *spatial grounding*. For instance, a user may provide not only a text prompt describing the image but also a set of bounding boxes indicating the desired positions of each object, as shown in Fig. 1. Such spatial constraints can be integrated into text-to-image (T2I) diffusion models by adding extra modules that are designed for spatial grounding and fine-tuning the model. GLIGEN [31] is a notable example, which incorporates a gated self-attention module [1] into the U-Net layers of Stable Diffusion [41]. Although effective, such fine-tuning-based approaches incur substantial training costs each time a new T2I model is introduced.

Recent training-free approaches for spatially grounded image generation [9, 47, 11, 36, 38, 48, 12, 26] have led to new advances, removing the high costs for fine-tuning. These methods leverage the fact that cross-attention maps in T2I diffusion models convey rich structural information about where each concept from the text prompt is being generated in the image [7, 19]. Building on this, these approaches aim to align the cross-attention maps of specific objects with the given spatial constraints (*e.g.* bounding boxes), ensuring that the objects are placed within their designated regions. This alignment is typically achieved by updating the noisy image in the reverse diffusion process using backpropagation from custom loss functions. However, such loss-guided update methods often struggle to provide precise spatial control over individual bounding boxes, leading to missing objects (Fig. 4, Row 9, Col. 5) or discrepancies between objects and their bounding boxes (Fig. 4, Row 4, Col. 4). This highlights the need for finer control over each bounding box during image generation.

We aim to provide more precise spatial control over each bounding box, addressing the limitations in previous loss-guided update approaches. A well-known technique for manipulating local regions of the noisy image during the reverse diffusion process is to directly replace or merge the pixels (or latents) in those regions. This simple but effective approach has proven effective in various tasks, including compositional generation [17, 50, 44, 32] and high-resolution generation [5, 29, 24, 25]. One could consider defining an additional branch for each bounding box, denoising with the corresponding text prompt, and then copying the noisy image into its designated area in the main image at each timestep. However, a key challenge lies in creating a noisy image patch–at the same noise level–that *reliably* contains the desired object while fitting within the specified bounding box. This has been impractical with existing T2I diffusion models, as they are trained on a limited set of image resolutions. While recent models such as PixArt-$\alpha$ [8] support a wider range of image resolutions, they remain constrained by specific candidate sizes, particularly for smaller image patches. As a result, when these models are used to create a local image patch, they are often limited to denoising a fixed-size image and cropping the region to fit the bounding box. This approach can critically fail to include the desired object within the cropped region.

In this work, we show that by exploiting the flexibility of the Transformer architecture, DiT can generate noisy image patches that fit the size of each bounding box, thereby reliably including each desired object. This is made possible through our proposed joint denoising technique. First, we introduce an intriguing property of DiT: when a smaller noisy patch is jointly denoised with a generatable-size noisy image, the two gradually become semantic clones—a phenomenon we call *semantic sharing*. Next, building on this observation, we propose a training-free framework that involves *cultivating* a noisy patch for each bounding box in a separate branch and then *transplanting* that patch into its corresponding region in the original noisy image. By iteratively transplanting the separately denoised patches into their respective bounding boxes, we achieved fine-grained spatial control over each bounding box region. This approach leads to more robust spatial grounding, particularly in cases where previous methods fail to accurately adhere to spatial constraints.

In our experiments on the HRS [3] and DrawBench [43] datasets, we evaluate our framework, GROUNDIT, using PixArt-$\alpha$ [8] as the base text-to-image DiT model. Our approach demonstrates superior performance in spatial grounding compared to previous training-free methods [38, 9, 47, 48], especially outperforming the state-of-the-art approach [47], highlighting its effectiveness in providing fine-grained spatial control.

## 2 Related Work

In this section, we review the two primary approaches for incorporating spatial controls into text-to-image (T2I) diffusion models: fine-tuning-based methods (Sec. 2.1) and training-free guidance techniques (Sec. 2.2).

### 2.1 Spatial Grounding via Fine-Tuning

Fine-tuning with additional modules is a powerful approach for enhancing T2I models with spatial grounding capabilities [51, 31, 2, 53, 46, 16, 10, 54]. SpaText [2] introduces a spatio-textual representation that combines segmentations and CLIP embeddings [40]. ControlNet [51] incorporates a trainable U-Net encoder that processes spatial conditions such as depth maps, sketches, and human keypoints, guiding image generation within the main U-Net branch. GLIGEN [31] enables T2I models to accept bounding boxes by inserting a gated attention module into Stable Diffusion [41]. GLIGEN's strong spatial accuracy has led to its integration into follow-up spatial grounding methods [48, 38, 30] and applications such as compositional generation [15] and video editing [23]. InstanceDiffusion [46] further incorporates conditioning modules to provide finer spatial control through diverse conditions like boxes, scribbles, and points. While these fine-tuning methods are effective, they require task-specific datasets and involve substantial costs, as they must be retrained for each new T2I model, underscoring the need for training-free alternatives.

### 2.2 Spatial Grounding via Training-Free Guidance

In response to the inefficiencies of fine-tuning, training-free approaches have been introduced to incorporate spatial grounding into T2I diffusion models. One approach involves a region-wise composition of noisy patches, each conditioned on a different text input [5, 50, 32]. These patches, extracted using binary masks, are intended to generate the object they are conditioned on within the generated image. However, since existing T2I diffusion models are limited to a fixed set of image resolutions, each patch cannot be treated as a complete image, making it uncertain whether the extracted patch will contain the desired object. Another approach leverages the distinct roles of attention modules in T2I models—self-attention captures long-range interactions between image features, while cross-attention links image features with text embeddings. By using spatial constraints such as bounding boxes or segmentation masks, spatial grounding can be achieved either by updating the noisy image using backpropagation based on a loss calculated from cross-attention maps [48, 38, 9, 7, 18, 36], or by directly manipulating cross- or self-attention maps to follow the given spatial layouts [26, 4, 14]. While the loss-guided methods enable spatial grounding in a training-free manner, they still lack precise control over individual bounding boxes, often leading to missing objects or misalignmet between objects and their bounding boxes. In this work, we propose a novel training-free framework that offers fine-grained spatial control over each bounding box by harnessing the flexibility of the Transformer architecture in DiT.

## 3 Background: Diffusion Transformers

Diffusion Transformer (DiT) [37] represents a new class of diffusion models that utilize the Transformer architecture [45] for their denoising network. Previous diffusion models like Stable Diffusion [41] use the U-Net [42] architecture, of which each layer contains a convolutional block and attention modules. In contrast, DiT consists of a sequence of DiT blocks, each containing a pointwise feedforward network and attention modules, removing convolution operations and instead processing image tokens directly through attention mechanisms.

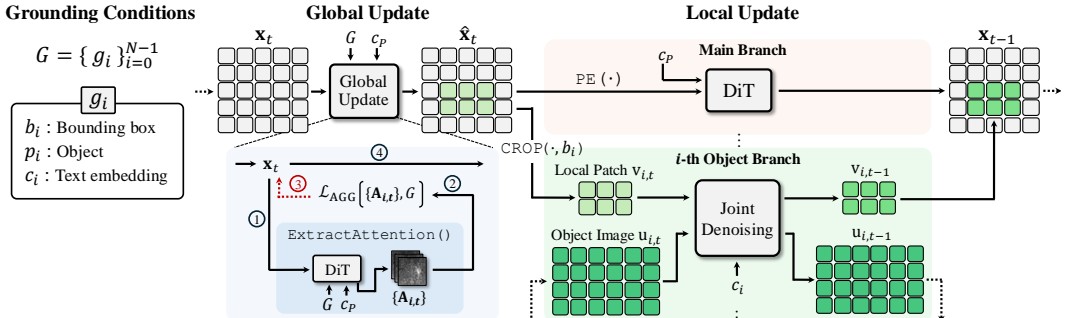

Figure 2: A single denoising step in GROUNDIT consists of two stages. The Global Update (Sec. 5.1) established coarse spatial grounding by updating the noisy image with a custom loss function. Then, the Local Update (Sec. 5.3) further provides fine-grained spatial control over individual bounding boxes through a novel technique called noisy patch transplantation.

DiT follows the formulation of diffusion models [22], in which the forward process applies noise to a real clean data $\mathbf{x}_0$ by

$$\mathbf{x}_t = \sqrt{\alpha_t}\mathbf{x}_0 + \sqrt{1 - \alpha_t}\epsilon \quad \text{where} \quad \epsilon \sim \mathcal{N}(0, I),\ \alpha_t \in [0, 1]. \tag{1}$$

The reverse process denoises the noisy data $\mathbf{x}_t$ through a Gaussian transition

$$p_\theta(\mathbf{x}_{t-1}|\mathbf{x}_t) = \mathcal{N}(\mathbf{x}_t; \mu_\theta(\mathbf{x}_t, t), \Sigma_\theta(\mathbf{x}_t, t)) \tag{2}$$

where $\mu_\theta(\mathbf{x}_t, t)$ is calculated by a learned neural network trained by minimizing the negative ELBO objective [27]. While $\Sigma_\theta(\mathbf{x}_t, t)$ can also be learned, it is usually set as time dependent constants.

**Positional Embeddings.** As DiT is based on the Transformer architecture, it treats the noisy image $\mathbf{x}_t \in \mathbb{R}^{h \times w \times d}$ as a set of image tokens. Specifically, $\mathbf{x}_t$ is divided into patches, each transformed into an image token via linear embedding. This results in $(h/l) \times (w/l)$ tokens, where $l$ is the patch size. Importantly, before each denoising step, 2D sine-cosine positional embeddings are assigned to each image token to provide spatial information as follows:

$$\mathbf{x}_{t-1} \leftarrow \texttt{Denoise}(\texttt{PE}(\mathbf{x}_t), t, c). \tag{3}$$

Here, $\texttt{PE}(\cdot)$ applies positional embeddings, $\texttt{Denoise}(\cdot)$ represents a single denoising step in DiT at timestep $t$, and $c$ is the text embedding. This contrasts with U-Net-based diffusion models, which typically do not utilize positional embeddings for the noisy image. Detailed formulations of the positional embeddings are provided in the **Appendix (Sec. A)**.

## 4 Problem Definition

Let $P$ be the input text prompt (*i.e.*, a list of words), which we refer to as the *global prompt*. Let $c_P$ be the text embedding of $P$. We define a set of $N$ grounding conditions $G = \{g_i\}_{i=0}^{N-1}$, where each condition specifies the coordinates of a bounding box and the desired object to be placed within it. Specifically, each condition $g_i := (b_i, p_i, c_i)$ consists of the following: $b_i \in \mathbb{R}^4$, the $xy$-coordinates of the bounding box's upper-left and lower-right corners, $p_i \in P$, the word in the global prompt describing the desired object within the box, and $c_i$, the text embedding of $p_i$. The objective is to generate an image that aligns with the global prompt $P$ while ensuring each specified object is accurately positioned within its corresponding bounding box $b_i$.

## 5 GROUNDIT: Grounding Diffusion Transformers

We propose GROUNDIT, a training-free framework based on DiT for generating images spatially grounded on bounding boxes. Each denoising step in GROUNDIT consists of two stages: Global Update and Local Update. Global Update ensures coarse alignment between the noisy image and the bounding boxes through a gradient descent update using cross-attention maps (Sec. 5.1). Following this, Local Update further provides fine-grained control over individual bounding boxes via a novel noisy patch transplantation technique (Sec. 5.3). This approach leverages our key observation of DiT's semantic sharing property, introduced in Sec. 5.2. An overview of this two-stage denoising step is provided in Fig. 2.

## 5.1 Global Update with Cross-Attention Maps

First, the noisy image $\mathbf{x}_t$ is updated to spatially align with the bounding box inputs. For this, we leverage the rich structural information encoded in cross-attention maps, as first demonstrated by Chefer *et al.* [7]. Each cross-attention map shows how a region of the noisy image correspond to a specific word in the global prompt $P$. Let DiT consist of $M$ sequential DiT blocks. As $\mathbf{x}_t$ passes through the $m$-th block, the cross-attention map $a_{i,t}^m \in \mathbb{R}^{h \times w \times 1}$ for object $p_i$ is extracted. For each grounding condition $g_i$, the *mean* cross-attention map $A_{i,t}$ is obtained by averaging $a_{i,t}^m$ over all $M$ blocks as follows:

$$A_{i,t} = \frac{1}{M} \sum_{m=0}^{M-1} a_{i,t}^m. \tag{4}$$

For convenience, we denote the operation of extracting $A_{i,t}$ for every $g_i \in G$ as below:

$$\{A_{i,t}\}_{i=0}^{N-1} \leftarrow \texttt{ExtractAttention}(\mathbf{x}_t, t, c_P, G). \tag{5}$$

Then, following prior works on U-Net-based diffusion models [48, 47, 38, 9, 36, 7], we measure the spatial alignment for each object $p_i$ by comparing its mean cross-attention map $A_{i,t}$ with its corresponding bounding box $b_i$, using a predefined grounding loss $\mathcal{L}(A_{i,t}, b_i)$ as defined in R&B [47]. The aggregated grounding loss $\mathcal{L}_{\text{AGG}}$ is then computed by summing the grounding loss across all grounding conditions $g_i \in G$:

$$\mathcal{L}_{\text{AGG}}(\{A_{i,t}\}_{i=0}^{N-1}, G) = \sum_{i=0}^{N-1} \mathcal{L}(A_{i,t}, b_i). \tag{6}$$

Based on the backpropagation from $\mathcal{L}_{\text{AGG}}$, $\mathbf{x}_t$ is updated via gradient descent as follows:

$$\hat{\mathbf{x}}_t \leftarrow \mathbf{x}_t - \omega_t \nabla_{\mathbf{x}_t} \mathcal{L}_{\text{AGG}} \tag{7}$$

where $\omega_t$ is a scalar weight value for gradient descent. We refer to Eq. 7 as the *Global Update*, as the whole noisy image $\mathbf{x}_t$ is updated based on an aggregated loss from all grounding conditions in $G$.

The Global Update achieves reasonable accuracy in spatial grounding. However, it often struggles with more complex grounding conditions. For instance, when $G$ contains multiple bounding boxes (*e.g.*, six boxes in Fig. 4, Row 9) or small, thin boxes (*e.g.*, Fig. 4, Row 5), the desired objects may be missing or misaligned with the boxes. These examples show that the Global Update lacks fine-grained, box-specific control, underscoring the need for precise controls for individual bounding boxes. In the following sections, we introduce a novel method to achieve this fine-grained spatial control.

## 5.2 Semantic Sharing in Diffusion Transformers

In this section, we present our observations on an intriguing property of DiT, *semantic sharing*, which will serve as a key building block for our main method in Sec. 5.3.

**Joint Denoising.** We observed that DiT can *jointly* denoise two different noisy images together. For example, consider two noisy images, $\mathbf{x}_t$ and $\mathbf{y}_t$, both at timestep $t$ in the reverse diffusion process. Position embeddings are applied to the image tokens according to their sizes, resulting in $\bar{\mathbf{x}}_t = \text{PE}(\mathbf{x}_t)$ and $\bar{\mathbf{y}}_t = \text{PE}(\mathbf{y}_t)$. Notably, the two noisy images can differ in size, providing flexibility in joint denoising. The key part is that the two noisy images—more precisely, two sets of image tokens—are merged into a single set $\mathbf{z}_t$. We denote this process as $\texttt{Merge}(\cdot)$ (see Alg. 1, line 4). $\mathbf{z}_t$ is passed through the DiT blocks, yielding the output $\mathbf{z}_{t-1}$, which is then split into the denoised versions $\mathbf{x}_{t-1}$ and $\mathbf{y}_{t-1}$ via $\texttt{Split}(\cdot)$. Joint denoising is illustrated in Fig. 3-(A), and a pseudocode for a single step of joint denoising is shown in Alg. 1.

**Semantic Sharing.** Surprisingly, we found that joint denoising of two noisy images generates semantically correlated content in corredping pixels, even when the initial random noise differs. Consider two noisy images, $\mathbf{x}_T \in \mathbb{R}^{h_\mathbf{x} \times w_\mathbf{x} \times d}$ and $\mathbf{y}_T \in \mathbb{R}^{h_\mathbf{y} \times w_\mathbf{y} \times d}$, both initialized from a unit Gaussian distribution $\mathcal{N}(0, I)$. We experiment with a reverse diffusion process in which, for the initial $100\gamma\%$ of the denoising steps ($\gamma \in [0, 1]$), $\mathbf{x}_T$ and $\mathbf{y}_T$ undergo joint denoising. For the remaining

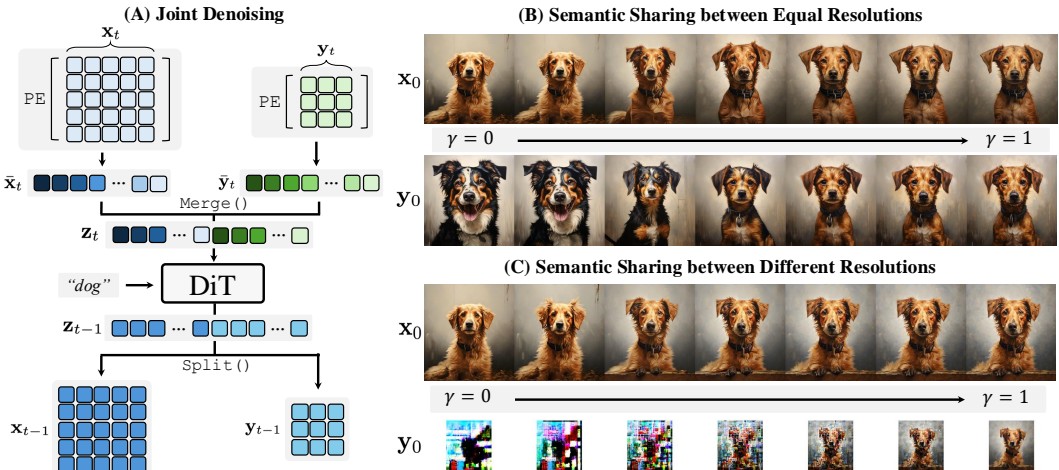

Figure 3: **(A) Joint Denoising.** Two different noisy images, $\mathbf{x}_t$ and $\mathbf{y}_t$, are each assigned positional embeddings based on their respective sizes. The two sets of image tokens are then merged and passed through DiT for a denoising step. Afterward, the denoised tokens are split back into $\mathbf{x}_{t-1}$ and $\mathbf{y}_{t-1}$. **(B), (C) Semantic Sharing.** Denoising two noisy images using joint denoising results in semantically correlated content between the generated images. Here, $\gamma$ indicates that joint denoising is during the initial $100\gamma\%$ of the timesteps, after which the images are denoised for the remaining steps.

timesteps, they are denoised independently. The same text embedding $c$ is used as a condition in both cases.

Fig. 3 shows the generated images from $\mathbf{x}_T$ and $\mathbf{y}_T$ across different $\gamma$ values. In Fig. 3-(B), $\mathbf{x}_T$ and $\mathbf{y}_T$ have the same resolution ($h_\mathbf{x} = h_\mathbf{y}, w_\mathbf{x} = w_\mathbf{y}$), while in Fig. 3-(C) their resolutions differ ($h_\mathbf{x} > h_\mathbf{y}, w_\mathbf{x} > w_\mathbf{y}$). When $\gamma = 0$, the two noisy images are denoised completely independently, resulting in clearly distinct images (leftmost column). We found that DiT models have a certain range of resolutions within which they can generate plausible images—which we refer to as *generatable resolutions*—but face challenges when generating images far outside this range. This is demonstrated in the output of $\mathbf{y}_0$ in Fig. 3-(C) with $\gamma = 0$. Further discussions and visual analyses are provided in the **Appendix (Sec. D)**. But as $\gamma$ increases, allowing $\mathbf{x}_T$ and $\mathbf{y}_T$ to be jointly denoised in the initial steps, the generated images become progressively more similar. When $\gamma = 1$, the images generated from $\mathbf{x}_T$ and $\mathbf{y}_T$ appear almost identical. These results demonstrate that, in joint denoising, assigning identical or similar positional embeddings to different image tokens promotes strong interactions between them during the denoising process. This correlated behavior during joint denoising causes the two image tokens to converge toward semantically similar outputs—a phenomenon we term *semantic sharing*.

Notably, this pattern holds not only when both noisy images share the same resolution (Fig. 3-(B)), but even when one of the images does not have DiT's generatable resolution (Fig. 3-(C)). While self-attention sharing techniques have been explored in U-Net-based diffusion models to enhance style consistency between images [20, 34], they have been limited to images of equal resolution. By leveraging the flexibility to assign positional embeddings across different resolutions, our joint

---

**Algorithm 1:** Pseudocode of Joint Denoising (Sec. 5.2).

**Inputs:** $\mathbf{x}_t \in \mathbb{R}^{h_\mathbf{x} \times w_\mathbf{x} \times d}, \mathbf{y}_t \in \mathbb{R}^{h_\mathbf{y} \times w_\mathbf{y} \times d}, t, c, l$; // Noisy images, timestep, text embedding, patch size.

**Outputs:** $\mathbf{x}_{t-1}, \mathbf{y}_{t-1}$;                                        // Noisy images at timestep $t-1$.

```
1 Function JointDenoise(x_t, y_t, t, c):
2     n_x ← h_x w_x / l²,  n_y ← h_y w_y / l²;        // Store the number of image tokens.
3     x̄_t ← PE(x_t),  ȳ_t ← PE(y_t);                    // Apply positional embeddings.
4     z_t ← Merge(x̄_t, ȳ_t);                       // Merge two sets of image tokens.
5     z_{t-1} ← Denoise(z_t, t, c);                      // Denoising step with DiT.
6     {x_{t-1}, y_{t-1}} ← Split(z_{t-1}, {n_x, n_y});        // Split back into two sets.
7     return x_{t-1}, y_{t-1};
```

denoising approach extends across heterogeneous resolutions, offering greater versatility. We provide further discussions and analyses on semantic sharing in the **Appendix (Sec. D)**.

### 5.3 Local Update with Noisy Patch Transplantation

In this section, we introduce our key technique: Local Update via noisy patch cultivation and transplantation. Building on DiT's semantic sharing property from Sec. 5.2, we show how this can be leveraged to provide precise spatial control over each bounding box.

**Main & Object Branches.** We propose a parallel denoising approach with multiple branches: one for the main noisy image $\hat{\mathbf{x}}_t$ and additional branches for each grounding condition $g_i$. The main branch denoises the main noisy image using the global prompt $P$, while each object branch is designed to denoise local regions within the bounding boxes, enabling fine-grained spatial control over each region. For each $i$-th object branch, there is a distinct *noisy object image* $\mathbf{u}_{i,t}$, initialized as $\mathbf{u}_{i,T} \sim \mathcal{N}(0, I)$. We predefine the resolution of the noisy object image $\mathbf{u}_{i,t}$ by searching in PixArt-$\alpha$'s generatble resolutions that closely match the aspect ratio of the corresponding bounding box $b_i$. With $\hat{\mathbf{x}}_t$ obtained from Global Update (Sec. 5.1), each branch performs denoising in parallel. Below we explain the denoising mechanism for each branch.

**Noisy Patch Cultivation.** In the main branch at timestep $t$, the noisy image $\hat{\mathbf{x}}_t$ is denoised using the global prompt $P$ as follows: $\tilde{\mathbf{x}}_{t-1} \leftarrow \texttt{Denoise}(\texttt{PE}(\hat{\mathbf{x}}_t), t, c_P)$, where $\hat{\mathbf{x}}_t$ is the output from the Global Update and $c_P$ is the text embedding of $P$. For the $i$-th object branch, there are two inputs: the noisy object image $\mathbf{u}_{i,t}$ and a subset $\mathbf{v}_{i,t}$ of image tokens extracted from $\hat{\mathbf{x}}_t$, corresponding to the bounding box $b_i$. We denote this extraction as $\mathbf{v}_{i,t} \leftarrow \texttt{Crop}(\hat{\mathbf{x}}_t, b_i)$, where $\mathbf{v}_{i,t} \in \mathbb{R}^{h_i \times w_i \times d}$ is referred to as a *noisy local patch*. Here, $h_i$ and $w_i$ corresponds to height and width of bounding box $b_i$, respectively. Joint denoising is then performed on $\mathbf{u}_{i,t}$ and $\mathbf{v}_{i,t}$ to yield their denoised versions:

$$\{\mathbf{u}_{i,t-1}, \mathbf{v}_{i,t-1}\} \leftarrow \texttt{JointDenoise}(\mathbf{u}_{i,t}, \mathbf{v}_{i,t}, t, c_i), \tag{8}$$

where $c_i$ is the text embedding of the object $p_i$.

Through semantic sharing with the noisy object image $\mathbf{u}_{i,t}$ during joint denoising, the denoised local patch $\mathbf{v}_{i,t-1}$ is expected to gain richer semantic features of object $p_i$ than it would without joint denoising. Note that even when the noisy local patch $\mathbf{v}_{i,t}$ does not meet the typical generatable resolution of DiT (since it often requires cropping small bounding box regions of $\hat{\mathbf{x}}_t$ to obtain $\mathbf{v}_{i,t}$), it offers a simple and effective way for enriching $\mathbf{v}_{i,t}$ of the semantic features of object $p_i$. We refer to this process as *noisy patch cultivation*.

**Noisy Patch Transplantation.** After cultivating local patches through joint denoising in Eq. 8, each patch is transplanted into $\tilde{\mathbf{x}}_{t-1}$, obtained from the main branch. The patches are transplanted in their original bounding box regions specified by $b_i$ as follows:

$$\tilde{\mathbf{x}}_{t-1} \leftarrow \tilde{\mathbf{x}}_{t-1} \odot (1 - \mathbf{m}_i) + \texttt{Uncrop}(\mathbf{v}_{i,t-1}, b_i) \odot \mathbf{m}_i \tag{9}$$

Here, $\odot$ denotes the Hadamard product, $\mathbf{m}_i$ is a binary mask for the bounding box $b_i$, and $\texttt{Uncrop}(\mathbf{v}_{i,t-1}, b_i)$ applies zero-padding to $\mathbf{v}_{i,t-1}$ to align its position with that of $b_i$. This transplantation enables fine-grained local control for the grounding condition $g_i$. After transplanting the outputs from all $N$ object branches, we obtain $\mathbf{x}_{t-1}$, representing the final output of GROUNDIT denoising step at timestep $t$. In $\mathbf{x}_{t-1}$, the image tokens within the $b_i$ region are expected to possess richer semantic information about object $p_i$ compared to the initial $\tilde{\mathbf{x}}_{t-1}$ from the main branch. This process is referred to as *noisy patch transplantation*. We provide implementation details and full pseudocode of a single GROUNDIT denoising step in the **Appendix (Sec. E).**

## 6 Results

In this section, we present the experiment results of our method, GROUNDIT, and provide comparisons with baselines. For the base text-to-image DiT model, we use PixArt-$\alpha$ [8], which builds on the original DiT architecture [37] by incorporating an additional cross-attention module to condition on text prompts.

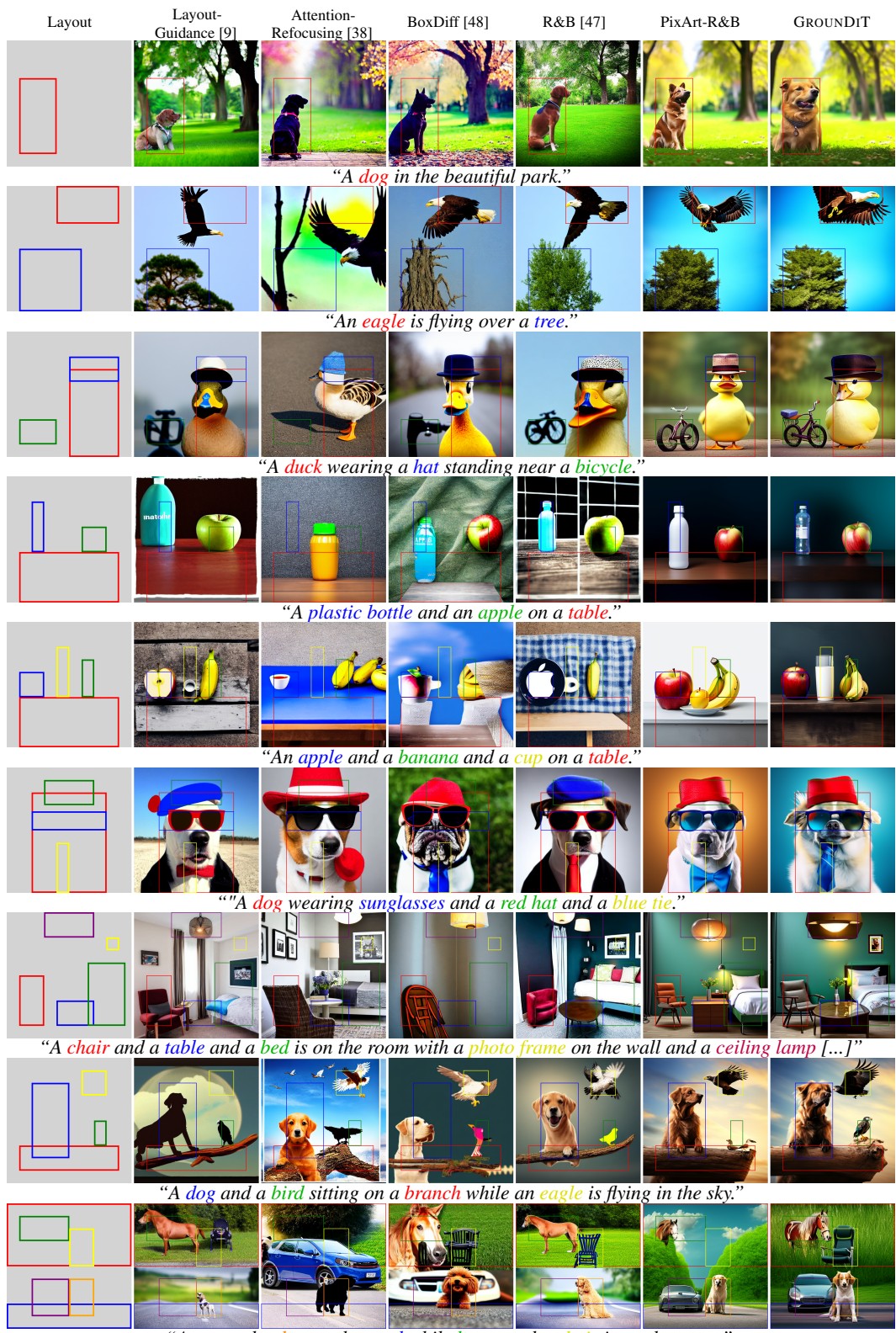

Figure 4: Qualitative comparisons between our GROUNDIT and baselines. Leftmost column shows the input bounding boxes, and columns 2-6 include the baseline results. The rightmost column includes the results of our GROUNDIT.

## 6.1 Evaluation Settings

**Baselines.** We compare our method with state-of-the-art training-free approaches for bounding box-based image generation, including R&B [47], BoxDiff [48], Attention-Refocusing [38], and Layout-Guidance [14]. For a fair comparison, we also implement R&B using PixArt-$\alpha$, which we refer to as *PixArt-R&B*, and treat it as an internal baseline. Note that this is identical to our method without the Local Guidance (Sec. 5.3).

**Evaluation Metrics and Benchmarks.**

- **(Grounding Accuracy)** We follow the evaluation protocol of R&B [47] to assess spatial grounding on the HRS [3] and DrawBench [43] datasets, using three criteria: spatial, size, and color. The HRS dataset consists of 1002, 501, and 501 images for each respective criterion, with bounding boxes generated using GPT-4 by Phung *et al.* [38]. For DrawBench, we use the same 20 positional prompts as in R&B [47].
- **(Prompt Fidelity)** We use the CLIP score [21] to evaluate how well the generated images adhere to the text prompt. Additionally, we assess our method using PickScore [28] and ImageReward [49], which provide human alignment scores based on the consistency between the text prompt and generated images.

| Method | HRS | | | DrawBench |
| | Spatial (%) | Size (%) | Color (%) | Spatial (%) |
| --- | --- | --- | --- | --- |
| **Backbone: Stable Diffusion [41]** | | | | |
| Stable Diffusion [41] | 8.48 | 9.18 | 12.61 | 12.50 |
| PixArt-$\alpha$ [8] | 17.86 | 11.82 | 19.10 | 20.00 |
| Layout-Guidance [9] | 16.47 | 12.38 | 14.39 | 36.50 |
| Attention-Refocusing [38] | 24.45 | 16.97 | 23.54 | 43.50 |
| BoxDiff [48] | 16.31 | 11.02 | 13.23 | 30.00 |
| R&B [47] | 30.14 | 26.74 | 32.04 | 55.00 |
| **Backbone: PixArt-$\alpha$ [8]** | | | | |
| PixArt-R&B | 37.13 | 20.76 | 29.07 | **60.00** |
| **GROUNDIT (Ours)** | **45.01** | **27.75** | **35.67** | **60.00** |

Table 1: Quantitative comparisons of grounding accuracy on HRS [3] and DrawBench [43] benchmarks. **Bold** represents the best, and underline represents the second best method.

## 6.2 Grounding Accuracy

**Quantitative Comparisons.** Tab. 1 presents a quantitative comparison of grounding accuracy between our method, GROUNDIT, and baselines. GROUNDIT outperforms all baselines across different criteria of grounding accuracy—spatial, size, and color—including the state-of-the-art R&B [47] and our internal baseline PixArt-R&B. Notably, the spatial accuracy on the HRS benchmark [3] (Col. 1) of GROUNDIT is significantly higher, with a +14.87% improvement over R&B and +7.88% over PixArt-$\alpha$. The comparison between PixArt-$\alpha$ [8], PixArt-R&B and GROUNDIT highlights the effectiveness of the two-stage pipeline of GROUNDIT. First, integrating the loss-based Global Update into PixArt-$\alpha$ results in a substantial improvement in spatial accuracy (from 17.86% to 37.13%). Then, incorporating our key contribution, the Local Update, further boosts accuracy (from 37.13% to 45.01%). For size accuracy (Col. 2), which evaluates how well the size of each generated object matches its corresponding bounding box, GROUNDIT shows a +1.01% improvement over R&B. In terms of color accuracy (Col. 3), our method achieves a +6.60% improvement over PixArt-R&B and outperforms R&B by +3.63%. This underscores the effectiveness of our noisy patch transplantation technique in accurately assigning color descriptions to the corresponding objects. As DrawBench [43] only contains images with two bounding boxes, which are relatively easy to generate, employing the Global Update is sufficient for grounding. We present additional quantitative comparisons of grounding accuracy in the **Appendix (Sec. B)**.

**Qualitative Comparisons.** Fig. 4 presents the qualitative comparisons. When the grounding condition involves one or two simple bounding boxes (Rows 1, 2), both our method and the baselines

successfully generate objects within the designated regions. However, as the number of bounding boxes increases and the grounding conditions become more challenging, the baselines struggle to correctly place each object inside the bounding box (Rows 4, 8), or even fail to generate the object at all (Rows 5, 7, 9). In contrast, GROUNDIT successfully grounds each object within the boxes, even when the number of boxes is relatively high, such as four boxes (Rows 5, 6, 8), five boxes (Row 7) and six boxes (Row 9). This highlights that our proposed noisy patch transplantation technique provides superior control over each bounding box, addressing the limitations of previous loss-based update methods, as discussed in Sec. 5.1. For more qualitative comparisons, including images generated with various aspect ratios, please refer to the **Appendix (Sec. G and Fig. 5)**.

## 6.3 Prompt Fidelity

Tab. 2 presents a quantitative comparison of prompt fidelity between our method and PixArt-R&B. Each metric is measured using the generated images from the HRS dataset [3]. GROUNDIT achieves higher CLIP score [21] than PixArt-R&B (Col. 1), indicating that our noisy patch transplantation improves the text prompt fidelity of the generated images. Additionally, our method achieves a higer ImageReward [49] score, which measures human preference by considering both prompt fidelity and overall image quality. While GROUNDIT shows a slight underperformance compared to PixArt-R&B in Pickscore [28], it remains comparable overall. We provide further comparisons of prompt fidelity with other baselines in the **Appendix (Sec. C)**.

| Method | CLIP score ↑ | ImageReward ↑ | PickScore ↑ |
|---|---|---|---|
| PixArt-R&B | 33.49 | 0.28 | **0.52** |
| **GROUNDIT (Ours)** | **33.63** | **0.44** | 0.48 |

Table 2: Quantitative comparisons on prompt fidelity on HRS benchmark [3]. **Bold** represents the best method.

## 7 Conclusion

In this work, we introduced GROUNDIT, a training-free spatial grounding technique for text-to-image generation, leveraging Diffusion Transformers (DiT). To address the limitation of prior approaches, which lacked fine-grained spatial control over individual bounding boxes, we proposed a novel approach that transplants a noisy patch generated in a separate denoising branch into the designated area of the noisy image. By exploiting an intriguing property of DiT, semantic sharing, which arises from the flexibility of the Transformer architecture and the use of positional embeddings, GROUNDIT generates a smaller patch by simultaneously denoising two noisy image: one with a smaller size and the other with a generatable resolution by DiT. Through semantic sharing, these two noisy images become semantic clones, enabling fine-grained spatial control for each bounding box. Our experiments on the HRS and DrawBench benchmarks demonstrated that GROUNDIT achieves state-of-the-art performance compared to previous training-free grounding methods.

**Limitations and Societal Impacts.** A limitation of our method is the increased computation time, as it requires a separate object branch for each bounding box. We provide further analysis on the computation time in the **Appendix (Sec. F)**. Additionally, like other generative AI techniques, our method is susceptible to misuse, such as creating deepfakes, which can raise significant concerns related to privacy, bias, and fairness. It is crucial to develop safeguards to control and mitigate these risks responsibly.

## Acknowledgements

We thank Juil Koo and Jaihoon Kim for valuable discussions on Diffusion Transformers. This work was supported by the NRF grant (RS-2023-00209723), IITP grants (RS-2019-II190075, RS-2022-II220594, RS-2023-00227592, RS-2024-00399817), and KEIT grant (RS-2024-00423625), all funded by the Korean government (MSIT and MOTIE), as well as grants from the DRB-KAIST SketchTheFuture Research Center, NAVER-Intel Co-Lab, Hyundai NGV, KT, and Samsung Electronics.

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

**Appendix**

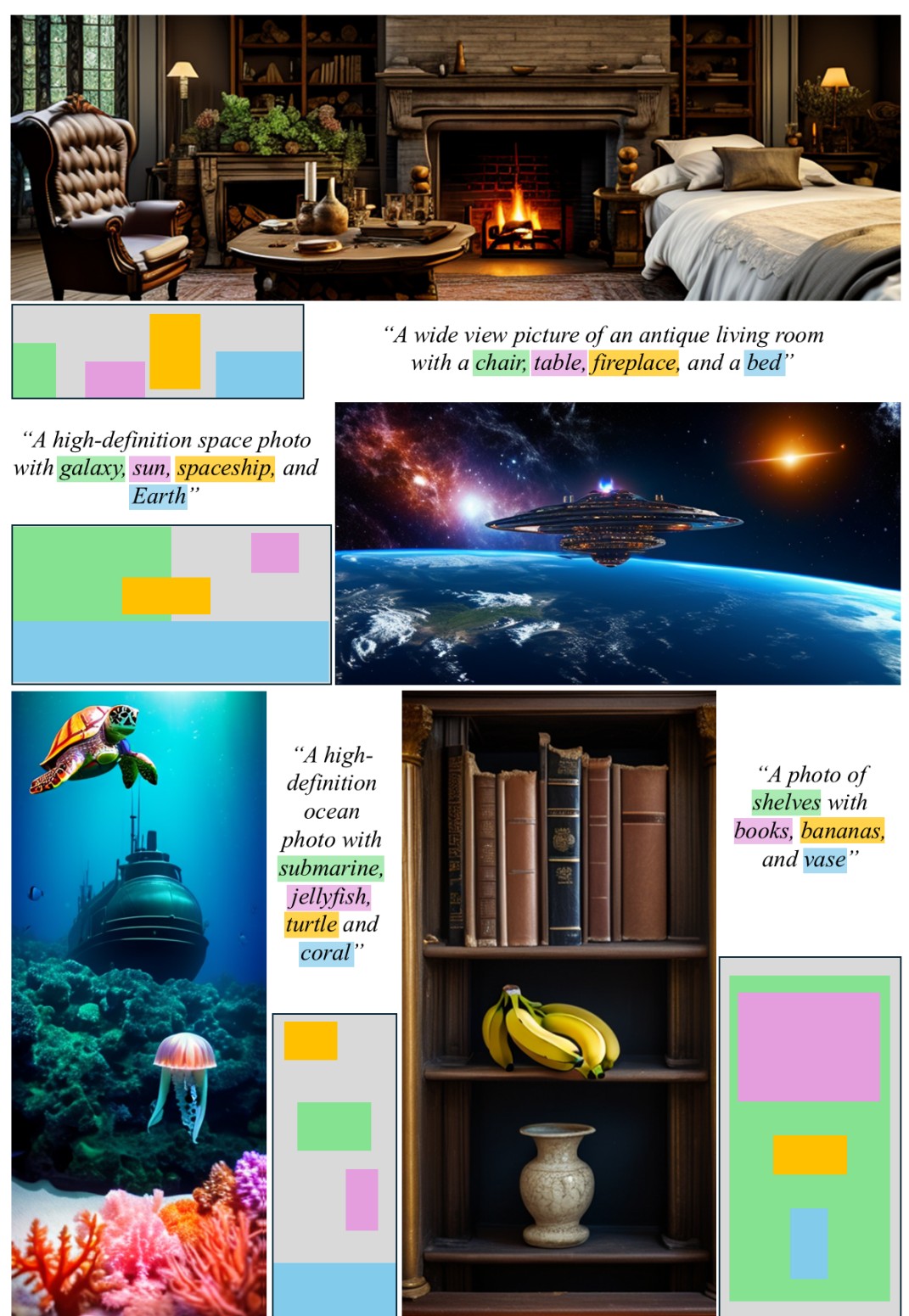

*"A wide view picture of an antique living room with a chair, table, fireplace, and a bed"*

*"A high-definition space photo with galaxy, sun, spaceship, and Earth"*

*"A high-definition ocean photo with submarine, jellyfish, turtle and coral"*

*"A photo of shelves with books, bananas, and vase"*

Figure 5: Spatially grounded images generated by our GROUNDIT with varying aspect ratios and sizes. Each image is generated based on a text prompt along with bounding boxes, which are displayed next to (or below) each image.

# A  Positional Embeddings in Diffusion Transformers

Diffusion Transformers (DiT) [37] handle noisy images of varying aspect ratios and resolutions by processing them as a set of image tokens. For this, the noisy image is first divided into patches, with each patch subsequently converted into an image token of hidden dimension $D$ through a linear embedding layer. DiT then applies 2D sine-cosine positional embeddings to each image token, based on its coordinates $(x, y)$, defined as follows:

$$p_{x,y} := \text{CONCAT}\,[p_x,\ p_y]\,, \quad \text{where} \quad p_x := [\cos(w_d \cdot x),\ \sin(w_d \cdot x)]_{d=0}^{D/4}$$
$$p_y := [\cos(w_d \cdot y),\ \sin(w_d \cdot y)]_{d=0}^{D/4}$$

where $w_d = 1/10000^{(4d/D)}$. The positional embedding $p_{x,y}$ is then added to each corresponding image token, denoted as $\text{PE}(\cdot)$.

# B  Additional Quantitative Comparisons: Grounding Accuracy

In addition to Sec. 6.2, we provide further quantitative comparisons of grounding accuracy between our GROUNDIT and the baselines. Specifically, we generated images based on text prompts and bounding boxes using each method, then calculated the mean Intersection over Union (mIoU) between the detected bounding boxes from an object detection model [55] and the input bounding boxes. Below, we present the quantitative comparisons across three datasets with varying average numbers of bounding boxes: subset of MS-COCO-2014 [33], HRS-Spatial [3], and a custom dataset.

| Dataset | Subset of MS-COCO-2014 [33] | HRS-Spatial [3] | Custom Dataset |
|---|---|---|---|
| Avg. # of Bounding Boxes | 2.06 | 3.11 | 4.48 |

Table 3: Average number of bounding boxes per dataset.

**Subset of MS-COCO-2014.**  We filtered the validation set of MS-COCO-2014 [33] to exclude image-caption pairs where the target objects were either not mentioned in the captions or duplicate objects were present. From this filtered set, we randomly selected 500 pairs for evaluation.

The results are presented in Tab. 4, column 2. GROUNDIT outperforms R&B by 0.021 (a 5.1% improvement) and PixArt-R&B by 0.014 (a 2.2% improvement). The relatively small margin can be attributed to the simplicity of the task, as this dataset has **an average of 2.06 bounding boxes** (Tab. 3), making it less challenging even for the baseline methods.

**HRS-Spatial.**  Column 3 of Tab. 4 presents the results on the *Spatial* subset of the HRS dataset [3]. GROUNDIT surpasses R&B [47] by 0.046 (a 14.1% improvement) and PixArt-R&B by 0.038 (an 11.4% improvement). Compared to the results on the MS-COCO-2014 subset, the higher percentage increase in mIoU highlights the robustness of GROUNDIT under more complex grounding conditions. Note that HRS-Spatial has **an average of 3.11 bounding boxes** (Tab. 3), which is higher than that of the MS-COCO-2014 subset (2.06).

**Custom Dataset.**  The custom dataset consists of 500 layout-text pairs, generated using the layout generation pipeline from LayoutGPT [15]. As shown in column 4 of Tab. 4, GROUNDIT outperforms R&B by 0.052 (a 26.3% improvement) and PixArt-R&B by 0.044 (a 21.4% improvement). This dataset has the **highest average number of bounding boxes at 4.48** (Tab. 3). These results further emphasize the robustness and effectiveness of our approach in handling more complex grounding conditions with a larger number of bounding boxes.

| Method | Subset of MS-COCO-2014 [33] | HRS-Spatial [3] | Custom Dataset |
|---|---|---|---|
| **Backbone: Stable Diffusion [41]** | | | |
| Stable Diffusion [41] | 0.176 | 0.068 | 0.030 |
| PixArt-$\alpha$ [8] | 0.233 | 0.085 | 0.036 |
| Layout-Guidance [9] | 0.307 | 0.199 | 0.122 |
| Attention-Refocusing [38] | 0.254 | 0.145 | 0.078 |
| BoxDiff [48] | 0.324 | 0.164 | 0.106 |
| R&B [47] | 0.411 | 0.326 | 0.198 |
| **Backbone: PixArt-$\alpha$ [8]** | | | |
| PixArt-R&B | 0.418 | 0.334 | 0.206 |
| **GROUNDIT (Ours)** | **0.432** | **0.372** | **0.250** |

Table 4: Quantitative comparisons of mIoU (↑) on a subset of MS-COCO-2014 [33], HRS-Spatial [3], and our custom dataset. **Bold** represents the best, and underline represents the second best method.

## C  Additional Quantitative Comparisons: Prompt Fidelity

In addition to Sec. 6.3, we provide further quantitative comparisons of the prompt fidelity of the generated images between our GROUNDIT and the baselines. We evaluated the generated images from the HRS dataset [3] using three different metrics: CLIP score [21], ImageReward [49], and PickScore [28]. The results are presented in Tab. 5. Since PickScore evaluates preferences between a pair of images, we report the difference between our GROUNDIT and each baseline method in column 4. Our GROUNDIT consistently outperforms the baselines in both CLIP score and ImageReward. For PickScore, GROUNDIT outperforms all baselines except PixArt-R&B, while remaining comparable.

| Method | CLIP score ↑ | ImageReward ↑ | PickScore ↑ (Ours − Baseline) |
|---|---|---|---|
| **Backbone: Stable Diffusion [41]** | | | |
| Layout-Guidance [9] | 32.48 | -0.401 | +0.30 |
| Attention-Refocusing [38] | 31.36 | -0.508 | +0.22 |
| BoxDiff [48] | 32.57 | -0.199 | +0.30 |
| R&B [47] | 33.16 | -0.021 | +0.26 |
| **Backbone: PixArt-$\alpha$ [8]** | | | |
| PixArt-R&B | 33.49 | 0.280 | -0.04 |
| **GROUNDIT (Ours)** | **33.63** | **0.444** | - |

Table 5: Quantitative comparisons of prompt fidelity on the HRS dataset [3]. **Bold** represents the best method.

## D  Additional Analysis on Semantic Sharing

In this section, we provide further analyses on the generatable resolution and the semantic sharing property of DiT, initially introduced in Sec. 5.2.

**Generatable Resolution of DiT.**  Although recent DiT models can generate images at various resolutions, they still struggle to produce images at *completely arbitrary* resolutions. We speculate that this limitation arises not from the model architecture itself, but from the resolution of the training images, which typically falls within a specific range [8]. Generating images at resolutions far outside this range often results in implausible outputs, suggesting the existence of an acceptable resolution range for DiT, which we refer to as its *generatable resolution*. In Fig. 6, we illustrate this phenomenon. When the noisy image size falls within DiT's generatable resolution range, the model produces plausible images (rightmost two images). However, when the image size is significantly outside this range (leftmost two images), DiT fails to generate a plausible image.

Prompt : " A dog"

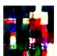 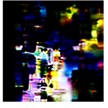 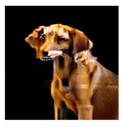 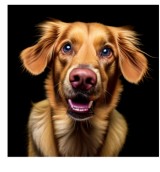 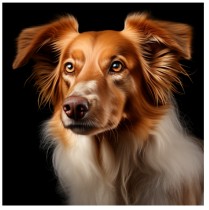

Image Size :  [128×128]          [256×256]          [288×288]          [384×384]                    [512×512]

Figure 6: Illustration of the generatable resolution range of DiT. The images are generated using PixArt-$\alpha$ [8] from the text prompt *"A dog"*, with varying resolutions.

**Semantic Sharing.**   Even though DiT models have a limited range of generatable resolutions, their Transformer architecture offers flexibility in handling varying lengths of image tokens, making it feasible to merge two sets of image tokens and denoise them through a single network evaluation. Leveraging this flexibility of Transformers, we presented our joint denoising technique (Alg. 1). Our main observation was that the joint denoising between two noisy images causes the two generated images to become semantically correlated, as illustrated in Fig. 3-(B) and Fig. 3-(C).

In addition to the visualizations in Fig. 3, we further quantify the semantic sharing property by measuring the LPIPS score [52] between two generated images. To explore the effect of joint denoising, we varied the parameter $\gamma \in [0, 1]$, which controls the proportion of denoising steps where joint denoising is applied. Specifically, $\gamma = 0$ means no joint denoising is applied, and each image is denoised independently, while $\gamma = 1$ means full joint denoising across all steps. As shown in Fig. 7, increasing $\gamma$ (*i.e.*, applying more joint denoising steps) results in a decrease in the LPIPS score between the two generated images, indicating that the images become more semantically similar as joint denoising is applied for a larger portion of the denoising process.

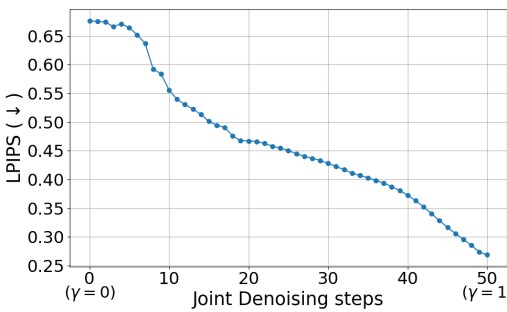 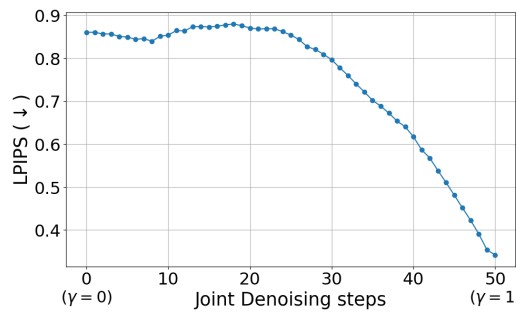

(a) Semantic sharing between equal resolutions          (b) Semantic sharing between different resolutions

Figure 7: LPIPS score between two generated images with varying $\gamma$ value. A gradual decrease in LPIPS [52] indicates that joint denoising progressively enhances the similarity between the generated images.

## E   Implementation Details

As the base text-to-image DiT model, we used the 512-resolution version of PixArt-$\alpha$ [8]. For sampling we employed the DPM-Solver scheduler [35] with 50 steps. Out of the 50 denoising steps, we applied our GROUNDIT denoising step (Alg. 2) for the initial 25 steps, and applied the vanilla denoising step for the remaining 25 steps. For the grounding loss in Global Update of GROUNDIT, we adopted the definition proposed in R&B [47], and we set the loss scale to 10 and used a gradient descent weight of 5 for the gradient descent update in Eq. 7.

As discussed in Sec. 5.3, for each $i$-th object branch we have a noisy object image $u_{i,t}$ and a noisy local patch $v_{i,t}$, which is extracted from the noisy image $\hat{\mathbf{x}}_t$ in main branch via $\mathbf{v}_{i,t} \leftarrow \texttt{Crop}(\hat{\mathbf{x}}_t, b_i)$. We determine the resolution of the noisy object image $u_{i,t}$ by selecting from PixArt-$\alpha$'s generatable resolutions, choosing one that best aligns with the aspect ratio of the corresponding bounding box $b_i$.

All our experiments were conducted an NVIDIA RTX 3090 GPU. In Algorithm 2, we provide the pseudocode of GROUNDiT single denoising step.

---

**Algorithm 2:** Pseudocode of GROUNDiT denoising step.

---

**Parameters :** $\omega_t$;                                                  // Gradient descent weight.
**Inputs:** $\mathbf{x}_t, \{\mathbf{u}_{i,t}\}_{i=0}^{N-1}, G, c_P$;     // Noisy images, grounding conditions, text embedding.
**Outputs:** $\mathbf{x}_{t-1}, \{\mathbf{u}_{i,t-1}\}_{i=0}^{N-1}$;                         // Noisy images at timestep $t-1$.

1 **Function** `GlobalUpdate`$(\mathbf{x}_t, t, c_P, G)$ **:**
     ┆  // $b_i$ holds coordinate information of bounding box, (Sec. 4)
2    ┆  $\{A_{i,t}\}_{i=0}^{N-1} \leftarrow \text{ExtractAttention}(\mathbf{x}_t, t, c_P, G)$;         // Extract cross-attention maps.
3    ┆  $\mathcal{L}_{\text{AGG}} \leftarrow \sum_{i=0}^{N-1} \mathcal{L}(A_{i,t}, b_i)$;             // Compute aggregated grounding loss.
4    ┆  $\hat{\mathbf{x}}_t \leftarrow \mathbf{x}_t - \omega_t \nabla_{\mathbf{x}_t} \mathcal{L}_{\text{AGG}}$;                             // Gradient descent (Eq. 7)
5    ┆  **return** $\hat{\mathbf{x}}_t$;

6 **Function** `LocalUpdate`$(\hat{\mathbf{x}}_t, \{\mathbf{u}_{i,t}\}_{i=0}^{N-1}, t, c_P, G)$ **:**
7    ┆  $\tilde{\mathbf{x}}_{t-1} \leftarrow \text{Denoise}(\hat{\mathbf{x}}_t, t, c_P)$;                                                // Main branch
8    ┆  **for** $i = 0, \ldots, N-1$ **do**
        ┆ ┆  // $i$-th object branch
9    ┆ ┆  $\mathbf{v}_{i,t} \leftarrow \text{Crop}(\hat{\mathbf{x}}_t, b_i)$;                                      // Obtain noisy local patch.
10   ┆ ┆  $\{\mathbf{u}_{i,t-1}, \mathbf{v}_{i,t-1}\} \leftarrow \text{JointDenoise}(\mathbf{u}_{i,t}, \mathbf{v}_{i,t}, t, c_i)$;         // Joint denoising.
11   ┆  **for** $i = 0, \ldots, N-1$ **do**
        ┆ ┆  // $\mathbf{m}_i$ is a binary mask corresponding to $b_i$
12   ┆ ┆  $\tilde{\mathbf{x}}_{t-1} \leftarrow \tilde{\mathbf{x}}_{t-1} \odot (1 - \mathbf{m}_i) + \text{Uncrop}(\mathbf{v}_{i,t-1}, b_i) \odot \mathbf{m}_i$;      // Patch Transplantation.
13   ┆  $\mathbf{x}_{t-1} \leftarrow \tilde{\mathbf{x}}_{t-1}$
14   ┆  **return** $\mathbf{x}_{t-1}, \{\mathbf{u}_{i,t-1}\}_{i=0}^{N-1}$;

15 **Function** `GroundDiTStep`$(\mathbf{x}_t, \{\mathbf{u}_{i,t}\}_{i=0}^{N-1}, t, c_P, G)$:
16   ┆  $\hat{\mathbf{x}}_t \leftarrow$ `GlobalUpdate`$(\mathbf{x}_t, t, c_P, G)$ ;                     // Global update (Sec. 5.1)
17   ┆  $\mathbf{x}_{t-1}, \{\mathbf{u}_{i,t-1}\}_{i=0}^{N-1} \leftarrow$ `LocalUpdate`$(\hat{\mathbf{x}}_t, \{\mathbf{u}_{i,t}\}_{i=0}^{N-1}, t, c_P, G)$ ;                     // Local update
     ┆  (Sec. 5.3)
18   ┆  **return** $\mathbf{x}_{t-1}, \{\mathbf{u}_{i,t-1}\}_{i=0}^{N-1}$;

---

## F   Analysis on Computation Time

We present the average inference time based on the number of bounding boxes in Tab. 6. While our method shows a slight increase in inference time, the rate of increase remains modest. For three bounding boxes, the inference time is 1.01 times that of R&B and 1.33 times that of PixArt-R&B. Even with six bounding boxes, the inference time is only 1.41 times that of R&B and 1.90 times that of PixArt-R&B.

| # of bounding boxes | 3 | 4 | 5 | 6 |
|---|---|---|---|---|
| R&B [47] | 37.52 | 38.96 | 39.03 | 39.15 |
| PixArt-R&B | 28.31 | 28.67 | 29.04 | 29.15 |
| GROUNDiT (Ours) | 37.71 | 41.10 | 47.83 | 55.30 |

Table 6: Comparison of the average inference time based on the number of bounding boxes. Values in the table are given in seconds.

## G   Additional Qualitative Results

We provide more qualitative comparisons in Fig. 8. Our method demonstrates greater robustness against issues such as the missing object problem, attribute leakage, or object interruption problem [47], due to its local update mechanism with semantic sharing. For instance, in Row 1, baseline methods struggle to generate certain objects (*i.e.* **missing object problem**). In Row 2, baselines generate a banana that retains features of an apple, illustrating **attribute leakage**. In Row 3, R&B generates a bus that interrupts the generation of a couch, with part of the bus overlapping with the designate region of the couch. Similarly, in PixArt-R&B, a hamburger and a donut interrupt the

| Layout | R&B [47] | PixArt-R&B | GROUNDiT |
|--------|----------|------------|----------|

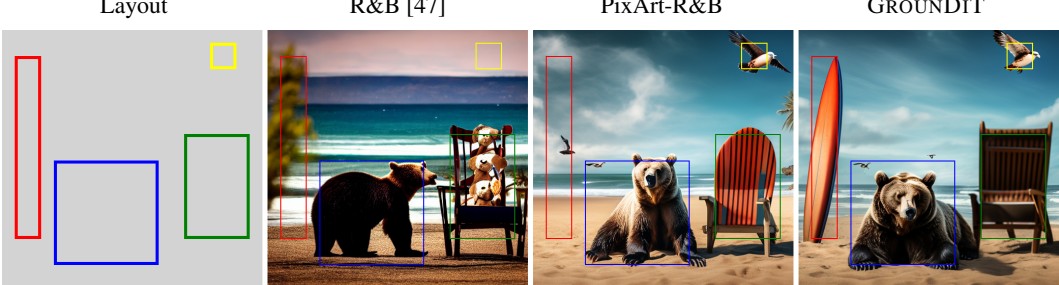

*"A bear sitting between a surfboard and a chair with a bird flying in the sky."*

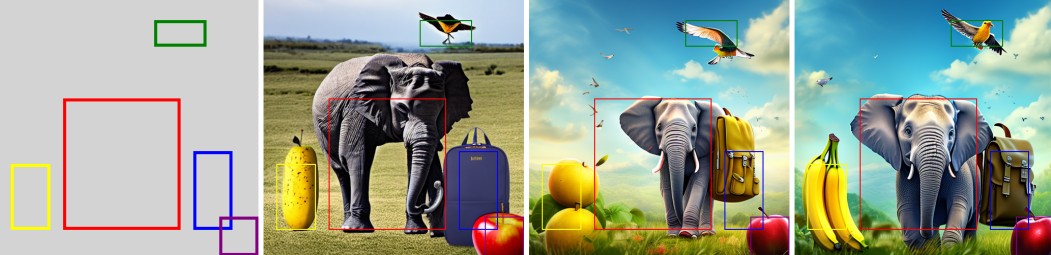

*"A banana and an apple and an elephant and a backpack in the meadow with bird flying in the sky."*

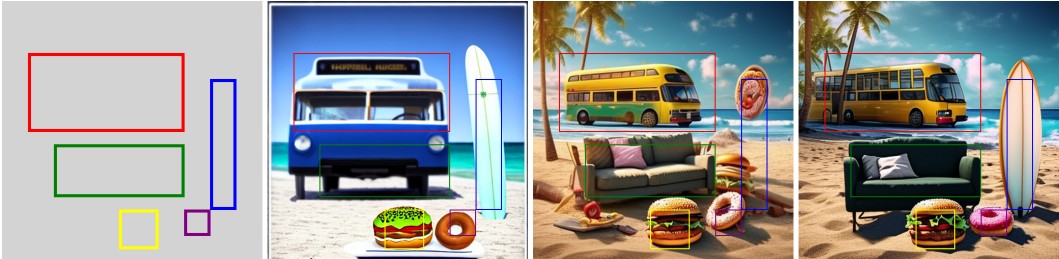

*"A realistic photo, a hamburger and a donut and a couch and a bus and a surfboard in the beach."*

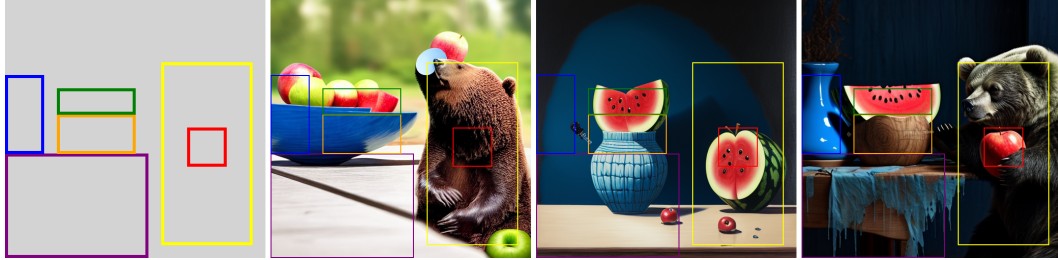

*"A blue vase and a wooden bowl with a watermelon sit on a table, while a bear holding an apple."*

Figure 8: Additional qualitative comparisons between our GROUNDiT, the previous state-of-the-art, R&B [47], and our internal baseline PixArt-R&B. Leftmost column shows the input bounding boxes, and columns 2-3 include the baseline results. The rightmost column includes the results of our GROUNDiT.

generation of a surfboard, demonstrating the **object interruption problem**. In more challenging cases, like Row 4, combinations of these issues appear. By contrast, our method consistently generates each object accurately within specified locations, even under complex bounding box configurations, highlighting its robustness and precision. Additional results are shown in Fig. 9, and examples of various aspect ratio images generated with grounding conditions are provided in Fig. 5.

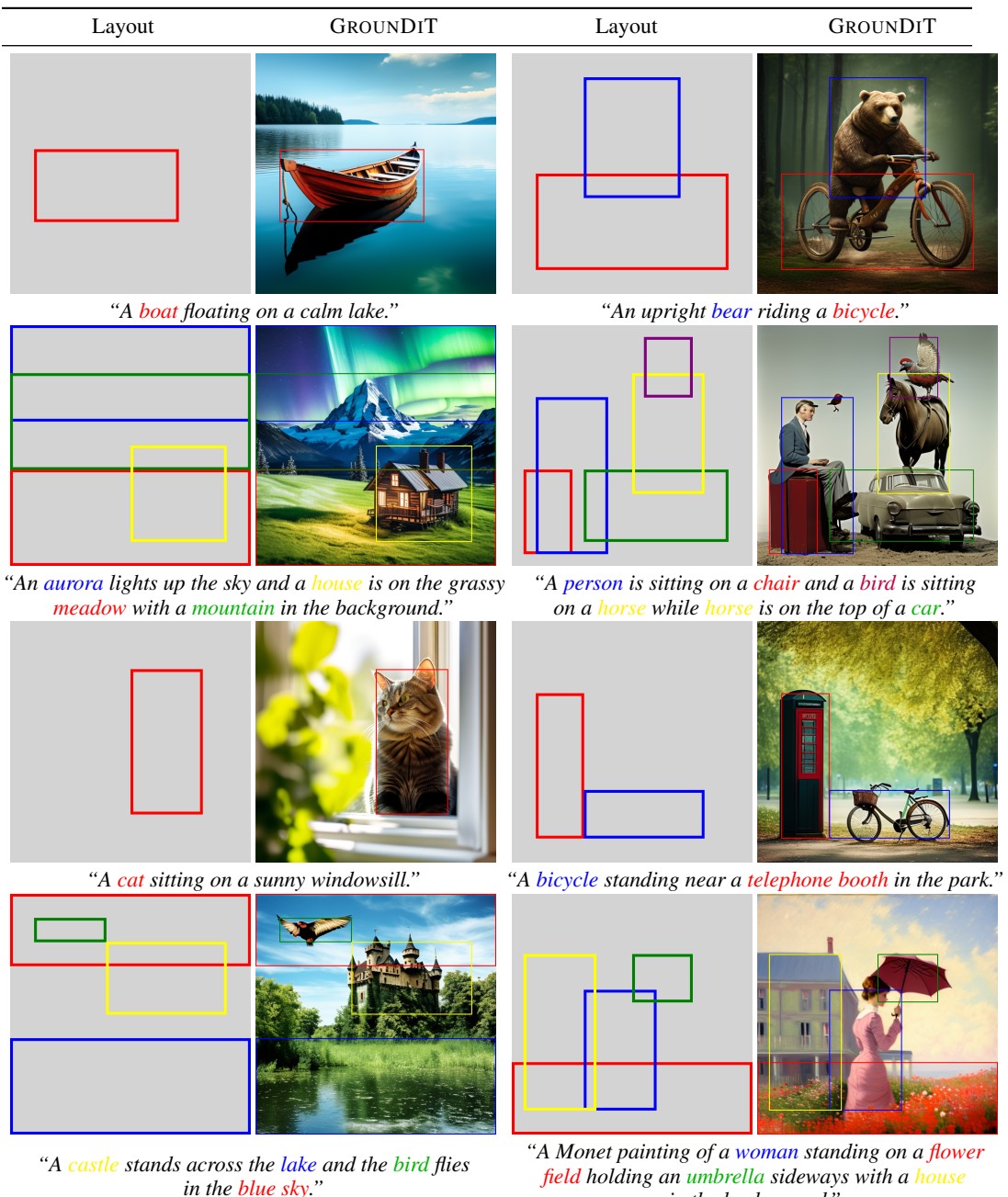

Figure 9: Additional spatially grounded images generated by out GROUNDIT.

