# OpenReview forum: "GrounDiT: Grounding Diffusion Transformers via Noisy Patch Transplantation"
_NeurIPS.cc/2024/Conference — NeurIPS 2024 poster_

### Official Review · Reviewer_xDuk · 2024-07-02

**Soundness:** 3
**Presentation:** 2
**Contribution:** 3
**Rating:** 6
**Confidence:** 4

**Summary:**

The paper introduces a zero-shot layout technique for spatially-grounded text-to-image generation with the power of transformer-based diffusion architecture. Prior methods tend to manipulate the latent image during reverse process for grounding, solely relying on cross-attention maps that show the alignment between specific text tokens and spatial regions of the target object to place in. This shows a drawback in generating a target object that fits within the region of bounding boxes. By utilizing transformer architecture, the method can divide the generation process into two branches: (1) first branch is to generate a target image patch of the bounding box with an arbitrary size by a transformer, (2) the second branch is the original generation process of the whole image. At every denoising step, the generated object is then copied into the generated image of the second branch.

**Strengths:**

- Addressing the limitation of attention-based method in generating arbitrary object sizes within bounding boxes, the method proposes to utilize transformer architecture (namely DiT) which is free from fixed-resolution constraints and enables patch-level generation.
- The paper is well-written and easy to follow.
- Model efficacy is superior to other methods in Figure 4 and Table 1.

**Weaknesses:**

- 158-161: It is obvious that different noise inputs certainly yield different outputs. So I do not understand the main point of this part.
- Semantic sharing part is an interesting observation. It is more helpful if the authors compare it with image interpolation. Does it work when two images have different semantic classes like dog and human?
Wang, Clinton, and Polina Golland. "Interpolating between images with diffusion models." (2023).
- Eq 5: What is formulation of $ \mathcal{L}(x_t, G_i, A_i)$?
- In reverse process: Have the authors applied semantic sharing on bounding boxes like concat tokens of multiple objects for simultaneous denoising? I assume that they share the same semantic class.
- In Fig 3, I saw the size of object image is much bigger than the size of bounding box. Is it intentional? If so, please explain. How to define noisy object image $z_{i, t}$? As far as I understand, the method need to reserve the aspect ratio of the bounding box by scaling it correspondingly to match the size of main image. Is it possible to disable this scaling step and denoise an object image with the same size as the bounding box?
- Does Noisy Patch transplantation guarantee the consistent content of the output image when pasting the generated image  with a binary mask (Eq. 9)? Again in Eq 9, the size of $x_{t-1}$ and $B_{i, t-1}$ are not the same as illustrated in Figure 3 so it is not clear how Hadamard product is possible.

**Questions:**

NA

**Limitations:**

Limitations and societal impacts are included in the paper.

---

> ### Author Rebuttal · Authors · 2024-08-06
>
> We sincerely appreciate your review, acknowledging our work to be “well-written” and possess “superior efficacy to other methods”. Here, we address the concerns and questions that have been raised.
>
> **(1) Clarification on the Main Method**
>
> We would like to draw your attention to our general response above, where we provide detailed clarifications regarding our problem definition and method. **Please note that the notations in the below sections are also based on the clarified versions.**
>
> **(2) Clarification on Lines 158-161 of Sec. 4.3**
>
> It is indeed obvious that if two noisy images $\mathbf{x}\_t$ and $\mathbf{y}\_t$ are initialized with different noises, they will produce different outputs if denoised separately (lines 158-161). However, we reiterate this fact to emphasize that as we increase the number of shared sampling between these two noisy images, $\mathbf{x}\_t$ and $\mathbf{y}\_t$ increasingly produce semantically similar outputs, even when starting from two different initial noises (lines 170-171). This phenomenon, also visualized in Fig. 2-(A), (B) of the main paper, signifies the intriguing effect of semantic sharing when two noisy images pass through the DiT together.
>
> **(3) Comparison between Semantic Sharing and Image Interpolation**
>
> Thank you for your comment on comparing our proposed method with the task of image interpolation. After reviewing the suggested paper [1], we would like to clarify the main purpose of our proposed shared sampling.
>
> Unlike image interpolation, which takes two (or more) input images and produces an output image with semantic features that lie between those of the input images, the goal of our shared sampling is to transfer the semantic features of one noisy image to the other. This process ensures that the desired concepts appear in the target noisy image.
>
> **(4) Clarification on Grounding Loss $\mathcal{L}$**
>
> Please note that we provide clarifications on the definition of the grounding loss in our general response. We will also include this clarification in the revised version.
>
> **(5) Clarification on Handling Multiple Objects of the Same Class**
>
> In Stage 2 of our GrounDiT, please note that a separate object branch and a corresponding noisy object image $\mathbf{z}\_{i, t}$ is defined and processed in parallel for each grounding condition $g_i$. This ensures that each object is individually handled, therefore providing precise local guidance for each bounding box region $b_i$.
>
> **(6) Further Clarification on Shared Sampling Mechanism**
>
> In Fig. 3 of the main paper, we intentionally depict the noisy object image $\mathbf{z}\_{i, t}$ as larger than the corresponding bounding box. This highlights that even when $\mathbf{z}\_{i, t}$ and the bounding box $b_i$ are not the same size, shared sampling between the two noisy images is still feasible. This is achieved by assigning positional embeddings so that each image is treated as a whole. This case is also visualized in Fig. 2-(B) of the main paper.
>
> Moreover, as describe in our general response, the noisy object image $\mathbf{z}\_{i,t}$ is set to satisfy the following criteria:
>
> * Size of $\mathbf{z}\_{i,t}\in\mathbb{R}^{H’_i \times W’_i \times D}$ is set within DiT’s preferred token sequence range.
> * $\mathbf{z}\_{i,t}$ has a similar aspect ratio as its corresponding bounding box $b_i$.
>
> Therefore, the noisy object image $\mathbf{z}\_{i, t}$ does not require a scaling process after it has been initialized. Simply passing $\mathbf{z}\_{i, t}$ and the nosy patch $\mathbf{b}\_{i,t}$ (originally $B_{i,t}$) together through the DiT for denoising is sufficient to perform shared sampling.
>
> **(7) Consistency of Images after Noisy Patch Transplantation**
>
> To prevent potential inconsistencies between the bounding box regions and the background, our GrounDiT guidance is applied during the initial timesteps of the reverse process, as detailed in Appendix A.1 of our main paper.
>
> Specifically, our reverse process consists of 50 steps, with GrounDiT denoising (Stage 1 and Stage 2) applied for the first 25 steps. For the final 25 steps, the noisy image $x_t$ is denoised using the standard DiT denoising step. This strategy, also employed in previous works such as R&B, leverages the fact that the image structure is primarily determined in the early steps. Therefore, applying our guidance in the early steps is sufficient for accurate grounding, while preventing inconsistencies between the bounding box regions and the background regions.
>
> **(8) Clarification on Hadamard Produce in Eq. (9)**
>
> Thank you for the comment. We correct Eq. (9) below. We will incorporate the clarification in our revised version.
>
> Let $\textbf{UNCROP}(\cdot)$ be a function that zero-pads a patch $\mathbf{b}_{i,t}$ at region $b_i$ to match the size of the original image. Then, the transplantation mechanism originally proposed in Eq. (9) can be clarified as follows:
>
> \begin{align}
> & \text{for}\\; i = 0, ..., N-1:\\\\
> & \quad\\; \tilde{\mathbf{x}}\_{t-1}\leftarrow \tilde{\mathbf{x}}\_{t-1}\odot (1-M_i)+ \textbf{UNCROP}(\mathbf{b}\_{i,t-1},b_i)\odot M_i\\\\
> & \mathbf{x}\_{t-1}\leftarrow \tilde{\mathbf{x}}\_{t-1}
> \end{align}
>
> Please note that the binary mask $M_i$ is passed through the Hadamard product with $\textbf{UNCROP}(\mathbf{b}\_{i,t-1},b_i)$, which has the same resolution as $M_i$.
>
> [1] Interpolating between Images with Diffusion Models, Wang et al., ICML 2024 Workshop

---

> > ### Comment · Reviewer_xDuk · 2024-08-12
> >
> > Thanks for the detailed response, it helps address my concerns. Most of my concerns are about the exposition and description of the method which is the main issue of the paper as can be seen in other reviewers. Apart from the writing, the additional benchmarks clearly demonstrate the effectiveness of the method against other baselines as seen from other reviewers. Hence, I still keep my initial score and vote for acceptance. Meanwhile, I encourage the authors to include the ablation of direct pasting object image of the object branch into noisy image of the main branch. This is a good evidence to support the claim in global response.

---

> > > ### Author Response · Authors · 2024-08-13
> > >
> > > Thank you sincerely for taking the time to review our submission. We greatly appreciate your valuable feedback and will work to improve our work based on your suggestions.

---

### Official Review · Reviewer_sEMS · 2024-07-07

**Soundness:** 2
**Presentation:** 1
**Contribution:** 3
**Rating:** 7
**Confidence:** 3

**Summary:**

The paper targets exploiting a pre-trained PixArt-alpha (a text-to-image multi-aspect diffusion transformer) to generate images conditioned on a set of text-labeled bounding boxes.

The authors start from the idea that, at inference time, a transformer can be given an arbitrary number of tokens, and that these tokens can have positional embedding corresponding to multiple images of multiple aspect ratios at once. Empirically, they observe that in this situation, the diffusion transformer is able to handle the generation process consistently and the outputs are multiple images with shared content but with different aspect ratios corresponding to the input PEs.

When generating the content of the bounding boxes, they take advange of this idea by simultaneously generating two version of the same bounding-box, one with positional embeddings corresponding to its location in the full image, and the other one with positional embeddings making the crop appear as a full image.
This ensures that the label is represented in the full-image version, and thanks to the shared content, also in the original box version, that can then be transplanted back into the image.

**Strengths:**

- The paper confirms an interesting property of multi-aspect diffusion transformers, in that tokens are able to interact at inference time, across combinations of positions that are not seen during training.
- They propose a clever way to exploit this capability in the context of a multi-aspect diffusion transformer.
- Handling spatial relations correctly is one of the major challenges that state-of-the-art text-to-image models have yet to overcome. The targetted task of grounding is very relevant as it is one way to circumvent the issue.
- The proposed solution obtains very good performance on the considered benchmark.

**Weaknesses:**

Mainly, I find that the exposition of the idea not very clear and could be improved:
- How the different branches, the two-stage process, and the diffusion scheduling interact is not clear:
  - Appendix A.1 reveals that Stage 1 and 2 are merged in a single full 50 steps denoising schedule, with Stage 1 only being used in the 25 first steps, which is not exactly in line with what is presented in the main paper. This information needs to be in the main paper, and ideally the presentation should be made more consistent with the implementation.
  - Figure 3 and Line 200 suggest that in the main branch x_t is denoised in parallel to the object branch, but it does not appear in Algorithm 1. This is inconsistent. And if the main brain does anything, it needs to be explained and detailed.
- Section 3 should present how multi-aspect/size is handled in diffusion transformers, as it is a key requirement for the proposed method. Also, Eq.3 introduces $w$ without definition.
- a smaller point that could be improved: in 4.2, while it can be derived from context and references that $\mathcal{L}$ needs to be a grounding loss, this is nowhere explicitly stated. If I strictly adhered to what is written in this paragraph, the authors could very well be arguing that any loss function that takes as input $(x_t, \mathcal(G)_i, \mathcal(A)_i)$ would be effective, which is obviously not what they are trying to say.

**Questions:**

I find this submission to be very interesting and show a lot of promise, but the description of the method is very confusing. For the discussion period, I hope the authors can provide the necessary clarifications and leave enough time for us to have a discussion together around their updated description.
Indeed, until the main points are convincingly clarified, succintly reviewed again, and we are confident that changes will be made, I cannot recommend acceptance. My main question is then:
- Can the authors clarify if anything happens in the main branch? Perhaps a full description of how the method works, including the reverse diffusion iterations would help.

Also, as a comparatively minor point, diffusion models are known for the flexibility of their sampling process. Considering how each stage and branch have compatible objectives, have the authors considered doing every update simultaneously in a single reverse diffusion process instead of multi-stage?

**Limitations:**

Limitations and impacts are discussed adequately.

---

> ### Author Rebuttal · Authors · 2024-08-06
>
> We greatly appreciate your review, recognizing our method to be “very interesting” and “show a lot of promise”. Here, we address the concerns and questions that have been raised.
>
> **(1) Clarifications on the Main Method**
>
> **Please find the detailed clarification of the method in our general response above.** We will thorougly improve our presentation in the revised version. Moreover, please note that the notations in the below sections are also based on the above clarification.
>
> **(2) Clarification on Grounding Loss $\mathcal{L}$**
>
> Please note that we provide clarifications on the definition of the grounding loss in our general response. We will also include this clarification in the revised version.
>
> **(3) Parallel Denoising of the Main Branch and Object Branch**
>
> While in Algorithm 1 we meant to convey the parallel process of main branch and object branches, we notice that the presentation can be improved for clarity. We would like to provide clarifications below and further improve in the revised version.
>
> **Main Branch:** First, Line 13 of Algorithm 1 indicates the denoising step of the noisy image in the main branch. Importantly, the output from this step, $\tilde{\mathbf{x}}\_{t-1}$ (originally $\mathbf{x}\_{t-1}$), is not directly used as an input for the denoising steps in each object branch.
>
> **Object Branch:** Lines 6-8 of Algorithm 1 represent the denoising steps of the object branches. Here, the output from the main branch, $\tilde{\mathbf{x}}\_{t-1}$, is not directly used as input for the denoising. Instead, the output noisy patch $\mathbf{b}\_{i, t}$ (originally $B_{i,t}$) in Line 8 is later pasted back into the output $\tilde{\mathbf{x}}\_{t-1}$ from the main branch, as shown in Line 9.
>
> Therefore, since the output from either the main branch or the object branches is not used as an input for the other branch, these processes are indeed parallel.
>
> **(4) Details on the Reverse Process of GrounDiT**
>
> We clarify the reverse process of GrounDiT, supplementing Appendix A.1 of the main paper.
>
> Our reverse process consists of 50 steps using the DPM-Solver scheduler, where the GrounDiT denoising (Stage 1 and Stage 2) is applied for the first 25 steps. For the final 25 steps, the noisy image $\mathbf{x}_t$ is denoised following the standard DiT denoising step. This strategy is also employed in previous works, including R&B, as the structure of the image is known to be mostly decided in the early steps. Therefore, applying our guidance in the early steps is sufficient for accurate grounding.
>
> **(5) Handling Multiple Image Sizes in DiT / Clarification on $w_d$ in Eq. (3)**
>
> We appreciate your feedback on enhancing the clarity of our descriptions in Sec. 3. To address this, we have clarified DiT's mechanism for handling multiple resolutions below and will incorporate this into our revised version.
>
> DiT [1] handles multi-aspect/size noisy images by operating on a sequences of tokens. Given a noisy image $\mathbf{x}\_{t}\in \mathbb{R}^{H\times W\times C}$, DiT first divides it into patches of size $l \times l$, creating a total of $T = (H/l) \times (W/l)$ patches. Each patch is then transformed into a token with a hidden dimension of $D$ through linear embedding. A 2D sine-cosine positional embedding is applied to the sequence of tokens. The detailed formulation of positional embedding is given in Eq. (3) of our main paper, where $w_d$ follows original definition of positional embedding:
> $w_d = 1/10000^{(4d/D)}$ with $d$ running from $0$ to $D/4$.
> Consequently, when images with different aspect ratios or sizes are input into DiT, the primary difference is the token sequence length, which is managed by assigning appropriate positional embeddings to each sequence.
>
> **(6) Simultaneous Update in a Single Reverse Diffusion Process**
>
> Please note that a single denoising step of our GrounDiT consists of the two-stage sequence: "Stage 1 $\rightarrow$ Stage 2". During Stage 1, a global update is applied to the noisy image $\mathbf{x}\_t$. In Stage 2, precise local control is provided for each bounding box $b_i$ using our proposed shared sampling technique. We found it reasonable to perform the shared sampling based on the output $\hat{\mathbf{x}}\_t$ obtained from Stage 1, as this allows to leverage the improved alignment resulting from the global update.
>
> [1] Scalable Diffusion Models with Transformers, Peebles et al., ICCV 2023

---

> > ### Comment · Reviewer_sEMS · 2024-08-12
> >
> > I have read the other reviews and rebuttal, and I thank the authors for all the clarifications to the algorithm.
> > While the initial submission lacked in terms of preciseness and clarity, I believe the content of the rebuttal fixes the main issues and demonstrate the willingness of the authors to update the submission accordingly.
> >
> > As such, standing by my initial assessment that the proposed method is a novel, clever, and inspiring solution to a relevant problem in text-to-image models, and now with no major unadressed concerns, I updated my evaluation and recommend acceptance.

---

> > > ### Author Response · Authors · 2024-08-12
> > >
> > > We sincerely appreciate your time and effort in reviewing our submission and rebuttal. Your valuable feedback is greatly appreciated, and we will work to improve the clarity of our work accordingly.

---

### Official Review · Reviewer_bA1A · 2024-07-14

**Soundness:** 3
**Presentation:** 3
**Contribution:** 3
**Rating:** 5
**Confidence:** 3

**Summary:**

This paper explores the zero-shot layout-to-image generation problem using bounding boxes and texts as conditional inputs. The authors propose a method called GrounDiT, which builds upon the recent Diffusion Transformers (DiT) model. Leveraging DiT's emergent property of semantic sharing, where two noisy images of different sizes can progressively converge during sampling, the authors introduce a key innovation: a separate object denoising branch for each bounding box condition, running in parallel to the main denoising branch. In each object branch, a noisy object patch is generated and then implanted into the original image. The authors compare GrounDiT with SOTA zero-shot methods on the HRS and DrawBench datasets, demonstrating that GrounDiT can generate complete objects in the correct regions while maintaining high overall image fidelity.

**Strengths:**

- The paper is well-written, with a clear and concise background and motivation.
- The idea of having a separate object branch for each bounding box is simple and intuitive, making the method easy to understand and implement.
- The authors conduct an extensive comparison with SOTA baselines, including R&B, BoxDiff, Attention-Refocusing, and Layout-Guidance, demonstrating the effectiveness of their approach.

**Weaknesses:**

- The paper lacks a comparison of prompt fidelity on the HRS dataset. Except for PixArt-R&B( a variant of GrounDiT), it would also be useful to include results from other baselines like R&B and BoxDiff to ensure the completeness of the work, even if they may use different backbones.

- In Table 1, GrounDiT outperforms its variant PixArt-R&B in grounding accuracy, which is expected given the special design of noisy object patch cultivation and transplantation. However, in Table 2, GrounDiT continue to surpass PixArt-R&B in prompt fidelity scores like CLIP score and ImageReward. I am curious to know where these improvements come from, as object patch cultivation and transplantation should only benefit grounding ability. Did the authors perform multiple runs for the experiments?

- The paper does not evaluate the MS-COCO subset presented in the R&B paper, which would be a valuable addition to the experiments.

- The proposed method would increase the compuation cost for image generation. However, there is no quantitative evidence showing the exact cost.

**Questions:**

Please refer to the weaknesses part.

**Limitations:**

In the conclusion part, the authors admit that the proposed method would increase the compuation time for image generation.

---

> ### Author Rebuttal · Authors · 2024-08-06
>
> We greatly appreciate your review, acknowledging that our paper is “well-written” and the proposed method is “simple and intuitive”. Here, we address the concerns and questions that have been raised.
>
> **(1) Additional Comparisons on Prompt Fidelity**
>
> Here we provide further quantitative comparisons on the prompt fidelity between our GrounDiT and baseline methods. We will include the results in the revised version.
>
> **[Prompt Fidelity]**
>
> | |Layout-Guidance|Attention-Refocusing|BoxDiff|R&B|PixArt-R&B|GrounDiT (Ours)|
> |:--|:---:|:---:|:---:|:---:|:---:|:---:|
> |CLIP score ($\uparrow$) | 32.48 | 31.36 | 32.57 | 33.16 | 33.49 | **33.63** |
> |ImageReward ($\uparrow$)| -0.401 | -0.508 | -0.199 | -0.021 | 0.280 | **0.444** |
> |PickScore (Ours $-$ Baseline)| +0.30 | +0.22 | +0.30 | +0.26 | -0.04 | - |
>
> The table above shows the comparison of CLIP score, ImageReward and PickScore with the baselines. Since PickScore measures the preference among two images, we present their difference in the above table. In CLIP score and ImageReward, our GrounDiT outperforms the baselines. In PickScore, GrounDiT outperforms all baseline except PixArt-R&B, while remaining comparable.
>
> **(2) Relation between Grounding and Prompt Fidelity**
>
> Additionally, we observed that precise local control provided by our GrounDiT method leads to the presence of target objects that are often missing in baseline methods, thereby increasing the prompt fidelity of the image. **Please refer to Fig. S1 in the attached PDF of our general response**, where we provide qualitative examples of these cases along with the corresponding CLIP scores for each image.
>
> **(3) Additional Quantitative Comparisons including MS-COCO**
>
> We further conducted quantitative comparisons on two additional benchmarks: a subset of the MS-COCO and a new custom benchmark.
>
> **[MS-COCO Subset]**
>
> Following R&B, we conducted an experiment on a subset of MS-COCO. Since we could not obtain the exact subset utilized in R&B from the authors, we similarly prepared a subset from MS-COCO.
>
> For this, we first filtered out the image-caption pairs in the MS-COCO 2014 validation set where either the bounding box target objects were not specified in the image captions, or duplicate objects were present in the grounding conditions. Subsequently, we randomly selected 500 pairs to use for evaluation. Following R&B, we measured the mIoU (mean IoU) for each bounding box condition.
>
> | |SD|PixArt-$\alpha$|Layout-Guidance|Attention-Refocusing|BoxDiff|R&B|PixArt-R&B|GrounDiT (Ours)|
> |:--|:--:|:---:|:---:|:---:|:---:|:---:|:---:|:---:|
> |mIoU ($\uparrow$) | 0.176 | 0.233 | 0.307 | 0.254 | 0.324 | 0.411 | 0.418 | **0.432** |
>
> As shown in the table above, our GrounDiT achieves the highest grounding accuracy on the MS-COCO subset. Specifically, GrounDiT outperforms R&B by 0.021, representing a 5.1% increase, and PixArt-R&B by 0.014, representing a 2.2% increase. Note that the average number of bounding boxes in the MS-COCO subset is **2.06**, making it a relatively easy task even for baseline methods. Since the main advantage of GrounDiT over R&B and PixArt-R&B is its robustness when there is a higher number of bounding boxes, we provide further comparisons with a higher average number of bounding boxes below.
>
> **[HRS-Spatial Benchmark]**
>
> | |SD|PixArt-$\alpha$|Layout-Guidance|Attention-Refocusing|BoxDiff|R&B|PixArt-R&B|GrounDiT (Ours)|
> |:--|:--:|:---:|:---:|:---:|:---:|:---:|:---:|:---:|
> |mIoU ($\uparrow$) | 0.068 | 0.085 | 0.199 | 0.145 | 0.164 | 0.326 | 0.334 | **0.372** |
>
> The above table provides the mIoU comparisons on the Spatial subset of the HRS benchmark, which was also used in Sec. 5.2. The average number of bounding boxes in this benchmark is **3.11**, which is larger than MS-COCO subset by 1.05. For mIoU, GrounDiT outperforms R&B by 0.046, indicating a 14.1% increase, and PixArt-R&B by 0.038, indicating a 11.4% increase. Compared to the 5.1% and 2.2% in the MS-COCO subset, respectively, the higher percentage of increase in mIoU demonstrates the robustness of GrounDiT on more complex grounding conditions.
>
> **[Custom Benchmark]**
>
> | |SD|PixArt-$\alpha$|Layout-Guidance|Attention-Refocusing|BoxDiff|R&B|PixArt-R&B|GrounDiT (Ours)|
> |:--|:--:|:---:|:---:|:---:|:---:|:---:|:---:|:---:|
> |mIoU ($\uparrow$) | 0.030 | 0.036 | 0.122 | 0.078 | 0.106 | 0.198 | 0.206 | **0.250** |
>
> Lastly, the above table shows the comparisons on a new benchmark consisting of 500 layout-text pairs, generated using the layout generation pipeline from LayoutGPT [1]. The average number of bounding boxes in this benchmark is **4.48**. Here, GrounDiT outperforms RnB by 0.052, representing a 26.3% increase and PixArt-R&B by 0.044 representing a 21.4% increase. This is in line with the trend in the above benchmarks, **further highlighting the robustness and efficacy of our approach in handling a higher number of grounding conditions.**
>
> **(4) Details on Computation Time**
>
> We further provide the exact computation time for each method.
>
> The numbers in the table below are in **seconds**.
>
> |Num. Boxes|3|4|5|6|
> |:--|:--:|:---:|:---:|:---:|
> |**R&B**| 37.52 | 38.96 | 39.03 | 39.15 |
> |**PixArt-R&B**| 28.31 | 28.67 | 29.04 | 29.15 |
> |**GrounDiT (Ours)**| 37.71 | 41.10 | 47.83 | 55.30 |
>
> While our method results in an increase in inference time, the rate of increase is not significant. For three bounding boxes, the inference time is 1.03 times that of R&B and 1.33 times that of PixArt-R&B for three bounding boxes. Even with 6 bounding boxes, the inference time is 1.41 times that of R&B and 1.90 times that of PixArt-R&B.
>
> [1] LayoutGPT: Compositional Visual Planning and Generation with Large Language Models, Feng et al., NeurIPS 2023

---

> > ### Comment · Reviewer_bA1A · 2024-08-14
> >
> > Thanks for the clarifications. After reading all the reponses and other reviewers' comments, I am inclined to keep my original score.

---

> > > ### Author Response · Authors · 2024-08-14
> > >
> > > Thank you for taking the time to review our submission and rebuttal. We greatly appreciate your valuable feedback and will work diligently to improve our work based on your insights.

---

### Official Review · Reviewer_LMpP · 2024-07-15

**Soundness:** 2
**Presentation:** 1
**Contribution:** 2
**Rating:** 4
**Confidence:** 4

**Summary:**

This paper proposes a method to use a pre-trained text-to-image
diffusion model to guide the generation into placing objects at given
locations determined by bounding boxes. The challenge is to develop
this capability without requiring fine-tuning of the model.

**Strengths:**

- The problem of guided generation without requiring fine-tuning of a
  diffusion model is a relevant one, and the method seems effective.
- Experimental results show the proposed method outperforms compared
  baselines.

**Weaknesses:**

- The quality of the presentation can be improved. For instance, one
  fundamental component of the method, $\math{L}_{AGG}$, is not
  defined. The evaluation metrics (spatial, size, color) are not
  properly defined, unless I missed it.
- Further analysis or discussion on why the proposed approach is
  superior to simply directly transferring the patch would make the
  paper more convincing.
- Experimental comparison is limited to two datasets. Qualitatively,
  it's not entirely clear there is a significant difference between
  the R&B baseline and the proposed approach.

**Questions:**

See weaknesses.

**Limitations:**

Yes

---

> ### Author Rebuttal · Authors · 2024-08-06
>
> We sincerely appreciate your review, acknowledging that our work is solving a “relevant” problem of zero-shot guided generation through an “effective” method. Here, we address the concerns and questions that have been raised.
>
> **(1) Clarifications on the Main Method**
>
> We would like to draw your attention to our general response above, where we provide detailed clarifications regarding our  method.
>
> **(2) Clarification on Grounding Loss $\mathcal{L}\_{AGG}$**
>
> Please note that we provide clarifications on the definition of the grounding loss in our general response. We will include this in the revised version.
>
> **(3) Details on Evaluation Metrics**
>
> We provide further details on the evaluation metrics from HRS-Bench [2]. We will also include this in the revised version.
>
> * **Spatial:** Object detection is applied to the generated image. Based on the detection results, the spatial relations specified in the text prompt are checked. If the detected objects maintain the correct spatial relations as described in the text prompt, the image is classified as correct; otherwise, it is classified as incorrect.
> * **Size:** Similarly, based on the detected boxes, it is checked whether the size relations between the boxes follow the relations specified in the text prompt (e.g. “bigger than”).
> * **Color:** After obtaining a semantic segmentation for each object, the average hue color value within each segment is computed. The derived color is then compared with the color of the object specified in the text prompt to determine accuracy.
>
> **(4) Quantitative Comparisons on Additional Benchmarks**
>
> We further conducted quantitative comparisons on two additional benchmarks: a subset of the MS-COCO and a new custom benchmark.
>
> **[MS-COCO Subset]**
>
> Following R&B, we conducted an experiment on a subset of MS-COCO. Since we could not obtain the exact subset utilized in R&B from the authors, we similarly prepared a subset of MS-COCO.
>
> We first filtered out the image-caption pairs in the MS-COCO 2014 validation set where either the bounding box target objects were not specified in the caption, or duplicate objects were present in the grounding conditions. Then we randomly selected 500 pairs for evaluation. Following R&B, we measured the mIoU (mean IoU) for each bounding box condition.
>
> | |SD|PixArt-$\alpha$|Layout-Guidance|Attention-Refocusing|BoxDiff|R&B|PixArt-R&B|GrounDiT (Ours)|
> |:--|:--:|:---:|:---:|:---:|:---:|:---:|:---:|:---:|
> |mIoU ($\uparrow$) | 0.176 | 0.233 | 0.307 | 0.254 | 0.324 | 0.411 | 0.418 | **0.432** |
>
> In the table above, our GrounDiT achieves the highest grounding accuracy on the MS-COCO subset. Specifically, GrounDiT outperforms R&B by 0.021, representing a 5.1% increase, and PixArt-R&B by 0.014, representing a 2.2% increase. Note that the average number of bounding boxes in the MS-COCO subset is **2.06**, making it a relatively easy task even for baseline methods. Since the main advantage of GrounDiT over R&B and PixArt-R&B is its robustness when there is a higher number of bounding boxes, we provide further comparisons with a higher average number of bounding boxes below.
>
> **[HRS-Spatial Benchmark]**
>
> | |SD|PixArt-$\alpha$|Layout-Guidance|Attention-Refocusing|BoxDiff|R&B|PixArt-R&B|GrounDiT (Ours)|
> |:--|:--:|:---:|:---:|:---:|:---:|:---:|:---:|:---:|
> |mIoU ($\uparrow$) | 0.068 | 0.085 | 0.199 | 0.145 | 0.164 | 0.326 | 0.334 | **0.372** |
>
> The above table provides the mIoU comparisons on the Spatial subset of the HRS benchmark, which was also used in Sec. 5.2. The average number of bounding boxes in this dataset is **3.11**, which is larger than MS-COCO subset by 1.05. For mIoU, GrounDiT outperforms R&B by 0.046, indicating a 14.1% increase, and PixArt-R&B by 0.038, indicating a 11.4% increase. Compared to the 5.1% and 2.2% in the MS-COCO subset, respectively, the higher percentage of increase in mIoU demonstrates the robustness of GrounDiT on more complex grounding conditions.
>
> **[Custom Benchmark]**
>
> | |SD|PixArt-$\alpha$|Layout-Guidance|Attention-Refocusing|BoxDiff|R&B|PixArt-R&B|GrounDiT (Ours)|
> |:--|:--:|:---:|:---:|:---:|:---:|:---:|:---:|:---:|
> |mIoU ($\uparrow$) | 0.030 | 0.036 | 0.122 | 0.078 | 0.106 | 0.198 | 0.206 | **0.250** |
>
> The above table shows the comparisons on a new benchmark consisting of 500 layout-text pairs, generated using the layout generation pipeline from LayoutGPT [1]. The average number of bounding boxes in this benchmark is **4.48**. Here, GrounDiT outperforms RnB by 0.052, representing a 26.3% increase and PixArt-R&B by 0.044 representing a 21.4% increase. This is in line with the trend in the above datasets, **further highlighting the robustness and efficacy of our approach in handling a higher number of grounding conditions.**
>
> **(5) Additional Qualitative Comparisons with R&B**
>
> Please refer to Fig. S1 of the attached PDF in the general response, where we provide additional qualitative comparisons between our method and the baselines, R&B and PixArt-R&B, with detailed analysis of each result.
>
> **(6) Advantage of Shared Sampling over Directly Transferring the Patches**
>
> Please refer to our general response for a clarification on the effect of shared sampling:
>
> We can think of each patch as an image on its own and pass it through DiT with the corresponding text condition. **However, this is not feasible in practice with existing DiT, as it has a certain set of preferred image resolutions.** This is due to the fact that the training data does not perfectly cover all resolutions. Please see Fig. S2 of the attached PDF in our general response for an illustration. But since DiT's Transformer architecture allows flexible token sequence length, we can perform shared sampling, which transfers the desired semantic features into the patch.
>
> [1] LayoutGPT: Compositional Visual Planning and Generation with Large Language Models, Feng et al., NeurIPS 2023
>
> [2] HRS-Bench: Holistic, Reliable and Scalable Benchmark for Text-to-Image Models, Bakr et al., ICCV 2023

---

### Author Rebuttal · Authors · 2024-08-06

Here we clarify our problem definition and method. For clarity, we have slightly modified the notations to improve the description of the entire framework and readability. We will continue to improve the presentation of our paper in the revision.

**[Notations]**

- Let $P$ be the input global text prompt.

* Grounding conditions are given as $G=[g_0,...,g_{N-1}]$. Each $g_i$ consists of a bounding box $b_i\in\mathbb{R}^4$ and a word $p_i$ indicating the desired object within $b_i$. **Note that $p_i$ must be one of the words in $P$**, that is, $p_i\in P$, since our method uses the cross-attention maps of DiT to compute the grounding loss $\mathcal{L}$ in Stage 1.

* Let $\mathbf{x}_t$ be a noisy image at timestep $t$. **Please note that DiT treats $\mathbf{x}_t$ as a sequence of tokens**, and applies positional embeddings at each timestep, as $\textbf{PE}(\mathbf{x}\_t)$.

**[Method Overview]**

Please note that our GrounDiT performs the reverse diffusion process **only once**, while conducting Stage 1 and 2 sequentially at every denoising step.

* **Stage 1** takes $\mathbf{x}_t$ and performs gradient descent to update $\mathbf{x}_t$ to $\hat{\mathbf{x}}_t$, which has not yet undergone a denoising step. $\hat{\mathbf{x}}_t$ is passed to Stage 2.

* **Stage 2** then denoises $\hat{\mathbf{x}}_t$ to obtain $\mathbf{x}\_{t-1}$, which moves to the next timestep of the reverse process.

**[Stage 1]**

We clarify Stage 1 as below:

* $\textbf{PE}(\mathbf{x}\_t)$ is passed into DiT, and in this process the cross-attention map $A_i$ for each word $p_i$ is extracted.

* With $A_i$, the grounding loss $\mathcal{L}(\mathbf{x}\_t,g_i,A_i)$ is computed. We use the loss functions in R&B [1], $\mathcal{L}=\mathcal{L}_r + \mathcal{L}_b$, where $\mathcal{L}_r$ (Eq. 11 in R&B) is a region-aware loss and $\mathcal{L}_b$ (Eq. 13 in R&B) is a region-aware loss.

* $\mathcal{L}\_{AGG}$ is computed by aggregating $\mathcal{L}$ for all conditions in $G$.

* Finally, we compute $\hat{\mathbf{x}}\_t \leftarrow \mathbf{x}\_t - \omega_t \nabla_{\mathbf{x}\_t} \mathcal{L}_{AGG}$.

While $\hat{\mathbf{x}}\_t$ is more likely to position each object of $p_i$ within the bounding box $b_i$, **it still struggles to accurately position every object in $b_i$.** Therefore, we introduce Stage 2, which focuses on a precise local update.

**[Stage 2]**

In Stage 2, we perform the denoising of $\hat{\mathbf{x}}\_t$ while injecting semantic features of object $p_i$ into the corresponding bounding box $b_i$.

For this, we define a single main branch as well as multiple object branches, each of which corresponds to a grounding condition $g_i\in G$, detailed below:

* **Main branch** is responsible for the denoising of main noisy image $\hat{\mathbf{x}}\_t$, **via the standard DiT denoising step** using the global prompt $P$:  $\tilde{\mathbf{x}}\_{t-1}\leftarrow \textbf{SampleDiT}(\textbf{PE}(\hat{\mathbf{x}}\_t), t, P)$. Note that $\tilde{\mathbf{x}}\_{t-1}$ is an intermediate denoised image, which is later combined with the outputs from object branches to finally obtain $\mathbf{x}\_{t-1}$.

* Each **object branch** is responsible for its corresponding grounding condition $g_i$.

   * Consider a local patch $\mathbf{b}\_{i,t}=\textbf{CROP}(\hat{\mathbf{x}}\_t, b_i)$. We want to ensure that the object $p_i$ appears within the bounding box $\mathbf{b}\_{i,t}$.

   * One can consider treating $\mathbf{b}\_{i,t}$ as an image on its own and passing it through DiT with $p_i$ as the text condition. **However, please note that this is not feasible with a pretrained DiT model since it cannot properly handle arbitrary resolutions of images.** This is NOT due to the architecture but to the fact that the training images typically have a certain resolution (which we call a **preferred resolution**). Please refer to Fig. S2 of the attached PDF.

   * However, please note that **DiT’s Transformer is still flexible to the length of its token sequence and thus feasible to combine two sequences of tokens**, *provided* that the positional embedding for each token sequence is computed with one of the preferred resolutions. Please see Fig 2 (A) in our paper, where both image has a preferred resolution.

   *  This leads us to our main idea: **shared sampling**. The noisy image $\mathbf{z}\_{i,t}$ at each branch has a preferred resolution (with a similar ratio to the bounding box $b_i$), while the patch $\mathbf{b}\_{i,t}$ cropped from the main branch does not. While we cannot directly denoise the patch $\mathbf{b}\_{i,t}$, we instead combine the two set of tokens from $\mathbf{z}\_{i,t}$ and $\mathbf{b}\_{i,t}$ and denoise them together while using $p_i$ as the text prompt. Our discovery is that this results in properly denoising the patch $\mathbf{b}\_{i,t}$ thanks to the interaction with the other sequence $\mathbf{z}\_{i,t}$, which has a preferred resolution, in the self-attention modules. Please see Fig 2 (B) in our paper. **Please note that $\mathbf{z}\_{i,t}$ in each object branch is NOT directly pasted onto the $\hat{\mathbf{x}}\_t$ in the main branch** but only used as an auxiliary to help properly denoise each patch in $\hat{\mathbf{x}}\_t$ with the corresponding object label $p_i$.

   * Specifically, we compute: $\\{ \mathbf{z}\_{i,t-1}, \mathbf{b}\_{i,t-1} \\}\leftarrow\textbf{SampleDiT}(\textbf{CONCAT}(\textbf{PE}(\mathbf{z}\_{i,t}),\textbf{PE}(\mathbf{b}\_{i,t})), t, p_i)$. Then, the denoised patch $\mathbf{b}\_{i,t-1}$ from $i$-th object branch is pasted back into its designated region in the main branch output $\tilde{\mathbf{x}}\_{t-1}$. The result of pasting back $\mathbf{b}\_{i,t-1}$ from all object branches into $\tilde{\mathbf{x}}\_{t-1}$ finally becomes $\mathbf{x}\_{t-1}$. The $\mathbf{x}\_{t-1}$ proceeds to the next timestep of the reverse process.

---

### Decision · Program_Chairs · 2024-09-25

**Decision:**

Accept (poster)

**Comment:**

This paper received mixed reviews: one accept, one weak accept, one borderline accept, and one borderline reject. The primary concerns raised by the reviewers were the precision and clarity of the method’s presentation, as well as the need for further comparisons on additional benchmarks.

The rebuttal effectively addressed these issues with additional evaluations and evidence, demonstrating the method’s effectiveness compared to existing approaches. Although reviewer LMpP was unable to respond to the rebuttal, the AC concurs with all the other reviewers that the improvements made in the rebuttal and subsequent discussions significantly raise the paper’s acceptance prospects.

The AC also agrees and thinks this work to be a novel, clear, and inspiring solution for text-to-image models, free from any noticeable flaws, and recommends it for acceptance.

For the camera-ready version, it is encouraged to incorporate the detailed responses from the rebuttal and discussions. Additionally, as suggested by reviewer xDuK, it would be beneficial to include an ablation study on the direct pasting of object images from the object branch into the noisy images of the main branch. We look forward to seeing this presentation at NeurIPS.